

# Model constraints on the anthropogenic carbon budget of the Arctic Ocean

**Jens Terhaar**[1], **James C. Orr**[1], **Marion Gehlen**[1], **Christian Ethé**[2], **and Laurent Bopp**[3]

[1]Laboratoire des Sciences du Climat et de l'Environnement, LSCE/IPSL, CEA-CNRS-UVSQ,
Université Paris-Saclay, Gif-sur-Yvette, France
[2]Institut Pierre et Simon Laplace, Paris, France
[3]LMD/IPSL, Ecole Normale Supérieure/PSL Research University, CNRS, Ecole Polytechnique,
Sorbonne Université, Paris, France

**Correspondence:** Jens Terhaar (jens.terhaar@lsce.ipsl.fr)

**Abstract.** The Arctic Ocean is projected to experience not only amplified climate change but also amplified ocean acidification. Modeling future acidification depends on our ability to simulate baseline conditions and changes over the industrial era. Such centennial-scale changes require a global model to account for exchange between the Arctic and surrounding regions. Yet the coarse resolution of typical global models may poorly resolve that exchange as well as critical features of Arctic Ocean circulation. Here we assess how simulations of Arctic Ocean storage of anthropogenic carbon ($C_{ant}$), the main driver of open-ocean acidification, differ when moving from coarse to eddy-admitting resolution in a global ocean-circulation–biogeochemistry model (Nucleus for European Modeling of the Ocean, NEMO; Pelagic Interactions Scheme for Carbon and Ecosystem Studies, PISCES). The Arctic's regional storage of $C_{ant}$ is enhanced as model resolution increases. While the coarse-resolution model configuration ORCA2 (2°) stores 2.0 Pg C in the Arctic Ocean between 1765 and 2005, the eddy-admitting versions ORCA05 and ORCA025 (1/2° and 1/4°) store 2.4 and 2.6 Pg C. The difference in inventory between model resolutions that is accounted for is only from their divergence after 1958, when ORCA2 and ORCA025 were initialized with output from the intermediate-resolution ORCA05. The difference would have been larger had all model resolutions been initialized in 1765 as was ORCA05. The ORCA25 Arctic $C_{ant}$ storage estimate of 2.6 Pg C should be considered a lower limit because that model generally underestimates observed CFC-12 concentrations. It reinforces the lower limit

from a previous data-based approach (2.5 to 3.3 Pg C). Independent of model resolution, there was roughly 3 times as much $C_{ant}$ that entered the Arctic Ocean through lateral transport than via the flux of $CO_2$ across the air–sea interface. Wider comparison to nine earth system models that participated in the Coupled Model Intercomparison Project Phase 5 (CMIP5) reveals much larger diversity of stored $C_{ant}$ and lateral transport. Only the CMIP5 models with higher lateral transport obtain $C_{ant}$ inventories that are close to the data-based estimates. Increasing resolution also enhances acidification, e.g., with greater shoaling of the Arctic's average depth of the aragonite saturation horizon during 1960–2012, from 50 m in ORCA2 to 210 m in ORCA025. Even higher model resolution would likely further improve such estimates, but its prohibitive costs also call for other more practical avenues for improvement, e.g., through model nesting, addition of coastal processes, and refinement of subgrid-scale parameterizations.

## 1 Introduction

The Arctic is experiencing amplified ocean acidification (Steinacher et al., 2009) and amplified climate change (Bekryaev et al., 2010), both of which may affect the marine ecosystem (Gattuso and Hansson, 2011). The main driver of the ongoing acidification of the open ocean is the increase in atmospheric $CO_2$ during the industrial era and the ensuing uptake of anthropogenic carbon from the atmosphere. Al-

though this absorbed anthropogenic carbon cannot be measured directly, being dominated by the natural component, it has been estimated from other oceanographic data.

For instance, Gruber et al. (1996) developed the $\Delta C^*$ method, building on seminal studies (Brewer, 1978; Chen and Millero, 1979) and their criticism (Broecker et al., 1985) as well as large new global data sets with improved $CO_2$ system measurements. That back-calculation method first calculates the total dissolved inorganic carbon ($C_T$) at equilibrium with the atmosphere before the water parcel is subducted. The preformed $C_T$ is then corrected for changes due to biological activity, as estimated from measurements of dissolved oxygen, total alkalinity ($A_T$), and nutrients, after which an estimate of preindustrial carbon is removed, finally yielding $\Delta C^*$. Yet the $\Delta C^*$ method's assumption of a constant air–sea $CO_2$ disequilibrium appears problematic in the high latitudes (Orr et al., 2001).

A second approach approximates the invasion of anthropogenic $CO_2$ into the interior ocean by a transient time distribution (TTD) method, itself constrained by observations of transient tracers such as CFC-12 or $SF_6$ (Hall et al., 2002; Waugh et al., 2004). A third approach uses Green's function instead of a TTD while also exploiting multiple transient tracers to assess the ocean's temporally changing distribution of anthropogenic carbon (Khatiwala et al., 2009). A comparison of these methods suggests that by 2010 the ocean had absorbed $155 \pm 31$ Pg C of anthropogenic carbon, around one-third of all emitted anthropogenic carbon (Khatiwala et al., 2013)

Less attention has been paid to anthropogenic carbon storage in the Arctic. Sabine et al. (2004) estimated that the Arctic Ocean had absorbed 4.9 Pg C by 1994. Yet without estimates for anthropogenic carbon in the Arctic itself, Sabine et al. (2004) scaled the Arctic inventory to be 5 % of their $\Delta C^*$-based estimate for global anthropogenic carbon storage, assuming the same Arctic : global ocean ratio as in the global gridded distribution of observed CFC-12 (Willey et al., 2004). More recently, Tanhua et al. (2009) used Arctic observations of CFC-11, CFC-12, and $SF_6$ and the TTD approach, revising the former Arctic anthropogenic carbon storage estimate downward to a range of 2.5 to 3.3 Pg C for year 2005. With that estimate, they emphasized that while the Arctic Ocean represents only 1 % of the global ocean volume, it stores 2 % of the global ocean's anthropogenic carbon. Although these numbers are relatively small, Arctic concentrations of anthropogenic $C_T$ must be relatively large, thus driving enhanced acidification in the Arctic Ocean. No other approaches have been used in the Arctic.

To provide an alternate approach to estimate anthropogenic carbon in the Arctic and to assess its budget and the mechanisms that control it, one could make carbon cycle simulations over the industrial era with a coupled ocean-circulation–biogeochemical model. A global-scale model configuration would be needed to account for the Arctic in the context of the global carbon cycle, while avoiding ar-

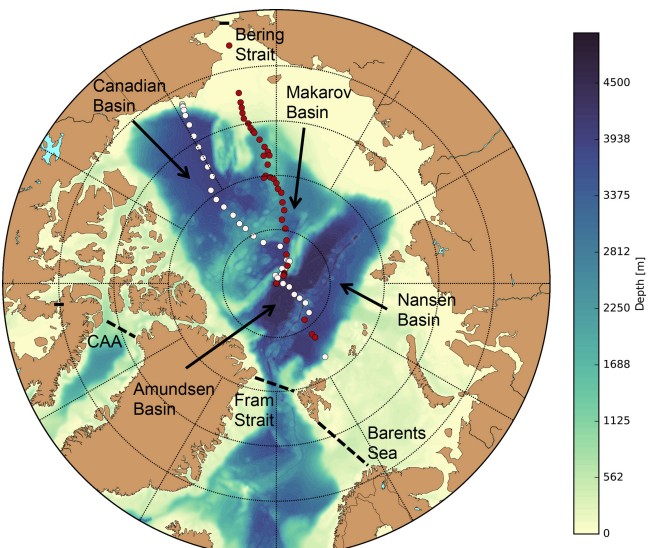

**Figure 1.** TS3 CFC-12 stations along the AOS94 (red) and Beringia 2005 expeditions (white). Other colors indicate the bathymetry of the Arctic Ocean, while the four dashed black lines show the boundaries of the Arctic Ocean domain used in this study.

tifacts from lateral boundary conditions that must be imposed in regional models. Yet typical ocean general circulation models have coarse resolution, which may be insufficient to adequately represent Arctic Ocean bathymetry, shelf, slopes, and ridges, all of which affect Arctic Ocean circulation (Rudels et al., 1994).

The bathymetry of the Arctic Ocean differs from that in other oceans in part because of the preponderance of shelf seas, comprising 53 % of the total surface area (Jakobsson, 2002) (Fig. 1). The remaining 47 % of its surface area covers 95 % of the total volume of the Arctic Ocean, split across four deep basins: the Nansen Basin, the Amundsen Basin, the Makarov Basin, and the Canadian Basin. Water masses enter these deep basins via (1) deep inflow from the Atlantic through the Fram Strait into the Nansen Basin, (2) inflow from the Barents Sea that sinks into the Nansen Basin through the St. Anna Trough, as cooling increases density, and (3) transport from density flows along the continental shelves that are driven by brine rejection from sea-ice formation (Jones et al., 1995). These three local transfers are difficult to resolve in coarse-resolution models, e.g., local density flows necessitate much higher resolution (Proshutinsky et al., 2016). Model resolution also affects the simulated interior circulation of the Arctic Ocean by its connection to the global ocean circulation via four relatively narrow and shallow passages: (1) the Canadian Archipelago, (2) the Fram Strait, (3) the Barents Sea Opening, and (4) the Bering Strait (Aksenov et al., 2016). Lateral exchange of water, carbon, and nutrients across these sections also affects Arctic Ocean primary production and acidification (Popova et al., 2013; Luo et al., 2016).

Here our aim is to use a three-dimensional model to help refine the estimate of the total anthropogenic carbon in the Arctic Ocean while assessing the dominant pathways by which anthropogenic carbon enters the Arctic Ocean and the relative importance of that lateral input relative to the air–sea flux. Three simulations made at increasingly higher grid resolution are aimed at assessing the extent to which the coarse resolution used by typical global ocean models may need to be improved to adequately estimate storage of anthropogenic carbon in the Arctic Ocean and associated ocean acidification.

## 2   Methods

Multiple global simulations were made to assess $C_{ant}$ in the ocean. Simulations were made with a state-of-the-art ocean-circulation–biogeochemical model at three resolutions over the industrial period since the mid-19th century, i.e., as is typical of recent model comparison efforts. Longer simulations were also made at the same resolutions with a less costly (and less precise) perturbation approach to correct for the missing anthropogenic carbon given that the actual industrial era began about a century earlier (mid-18th century). The highest-resolution configuration used an unprecedented lateral grid spacing for such long, global, biogeochemical simulations, although its cost meant its effect could be assessed only over 1958–2012.

### 2.1   Models

For our study, we used the global ocean circulation model NEMO-v3.2 (Nucleus for European Modeling of the Ocean – version 3.2). The NEMO model has three parts: (1) the ocean dynamics and thermodynamics model OPA CE2 (Madec, 2008), (2) the sea-ice model LIM CE3 (Vancoppenolle et al., 2009), and (3) the passive tracer module TOP. This physical model is coupled via TOP to version 1 of PISCES (Pelagic Interactions Scheme for Carbon and Ecosystem Studies) (Aumont and Bopp, 2006). For this study we used NEMO at three resolutions: a laminar 2° configuration (ORCA2) typical of coarse-resolution ocean models (Madec et al., 1998), which does not resolve eddies; an intermediate 0.5° configuration (ORCA05) that just begins to allow eddies to appear spontaneously (Bourgeois et al., 2016); and a higher-resolution, eddy-admitting version, i.e., a 0.25° configuration (ORCA025), which is still not eddy resolving (Barnier et al., 2006). The highest-resolution simulation is referred to as ORCA025-G70 in the DRAKKAR CE4 ensemble.

All three configurations have a tripolar, curvilinear horizontal grid. One grid pole (singularity) is located at the geographical South Pole while the conventional North Pole grid singularity over the Arctic Ocean has been replaced by two grid singularities, both displaced over land – one over Canada and the other over Russia (Madec et al., 1998) – thereby

saving computational costs and avoiding numerical artifacts. From 90° S to 20° N, the grid is a normal Mercator grid; north of 20° N, it is distorted into ellipses to create the two northern singularities (Barnier et al., 2006; Madec, 2008). The grid size changes depending on resolution and location (Table 1). The mean horizontal grid size in the Arctic Ocean (average length of the four horizontal edges of surface grid cells in the Arctic Ocean) is 121 km in ORCA2, 29 km in ORCA05, and 14 km in ORCA025. The smallest horizontal grid size in the Arctic is 63 km in ORCA2, 9 km in ORCA05, and 3 km in ORCA025.

Vertically, all three model configurations have the same discretization, where the full-depth water column is divided into 46 levels whose thicknesses vary from 6 m (top level) to 249 m (level 45), but the latter can reach up to 498 m, being extended into level 46 as a function of the bathymetry (partial steps). For its bathymetry, the ocean model relies on the 2 min bathymetry file ETOPO2 from the National Geophysical Data Center, which is based on satellite-derived data (Smith and Sandwell, 1997) except for the highest latitudes: the IBCAO CE5 bathymetric data are used in the Arctic (Jakobsson et al., 2000) and BEDMAP CE6 bathymetric data are used for the Southern Ocean south of 72° S (Lythe and Vaughan, 2001). To interpolate the bathymetry on the model grid, the median of all data points in one model grid cell was computed. NEMO uses the partial-step approach for the model to better match the observed topography. In this approach, the bathymetry of the model is not tied directly to the bottom edge of the deepest ocean grid level, which varies with latitude and longitude; rather, the deepest ocean grid level for each column of grid cells is partially filled in to better match the observed ocean bathymetry (Barnier et al., 2006).

The lateral isopycnal diffusion and viscosity coefficients were chosen depending on the resolution (Table 2). In ORCA2, a Laplacian viscosity operator was used, whereas a bi-Laplacian operator was used in ORCA05 and ORCA025. To simulate the effect of eddies on the mean advective transport in the two coarser-resolution configurations, the eddy parameterization scheme of Gent and Mcwilliams (1990) was applied with eddy diffusion coefficients indicated in Table 2. Vertically, the same eddy viscosity ($1.2 \times 10^{-4}\ \mathrm{m^2\,s^{-1}}$) and diffusivity coefficients ($1.2 \times 10^{-5}\ \mathrm{m^2\,s^{-1}}$) were used in all three resolutions.

The biogeochemical model PISCES (Aumont and Bopp, 2006) includes four plankton functional types: two phytoplankton types (nanophytoplankton and diatoms) and two zooplankton types (micro- and meso-zooplankton). The growth of phytoplankton is limited by the availability of five nutrients: nitrate, ammonium, total dissolved inorganic phosphorus $P_T$, total dissolved silicon $Si_T$, and iron. The nanophytoplankton and diatoms are distinguished by their need for all nutrients, with only diatoms requiring silicon. While the Fe : C and Chl : C ratios of both phytoplankton groups as well as the Si : C ratio of diatoms are predicted

**Table 1.** Grid size in the Arctic Ocean and volumes by basin as a function of model resolution.

| Configuration | Horizontal grid (km) | | | Volume ($10^6$ km$^3$) | | | | |
| --- | --- | --- | --- | --- | --- | --- | --- | --- |
| | | | | | Basins | | | |
| | Mean | Min | Max | Arctic | Nansen | Amundsen | Makarov | Canada |
| ORCA2 | 120.8 | 63.3 | 180.5 | 14.3 | 2.8 | 3.2 | 2.2 | 4.7 |
| ORCA05 | 29.0 | 9.4 | 41.3 | 13.3 | 2.6 | 2.7 | 1.9 | 4.9 |
| ORCA025 | 14.4 | 3.2 | 20.5 | 13.3 | 2.3 | 2.9 | 1.8 | 5.0 |

**Table 2.** Coefficients for lateral diffusivity, lateral viscosity, and eddy-induced velocity for ORCA2, ORCA05, and ORCA025.

| Configuration | Lateral diffusivity | Lateral viscosity[a] | Eddy-induced velocity |
| --- | --- | --- | --- |
| ORCA2[b] | 2000 m$^2$ s$^{-1}$ | $4 \times 10^4$ m$^2$ s$^{-1}$[c] | 2000 m$^2$ s$^{-1}$ |
| ORCA05 | 600 m$^2$ s$^{-1}$ | $-4 \times 10^{11}$ m$^2$ s$^{-1}$ | 1000 m$^2$ s$^{-1}$ |
| ORCA025 | 300 m$^2$ s$^{-1}$ | $-1.5 \times 10^{11}$ m$^2$ s$^{-1}$ | none |

[a] In ORCA2, a Laplacian viscosity operator was used, whereas a bi-Laplacian operator was used in ORCA05 and ORCA025. [b] Lateral diffusivity and viscosity coefficients decrease towards the poles proportional to the grid size. [c] Reduced to 2100 m$^2$ s$^{-1}$ in the tropics (except along western boundaries).

prognostically by PISCES, the remaining macronutrient ratios are held constant at $C : N : P = 122 : 16 : 1$ (Takahashi et al., 1985). The same ratio holds for nonliving compartments: dissolved organic matter (DOM) and both small and large sinking particles, which differ in their sinking velocity. In PISCES, nutrients are supplied by three external pathways: atmospheric dust deposition, river delivery, and sediment mobilization of iron. Dust deposition was taken from a simulation by Tegen and Fung (1995). River discharge of $C_T$ and dissolved organic carbon (DOC) is based on the Global Erosion Model (GEM) by Ludwig et al. (1998). Riverine DOC was assumed to be entirely labile, being instantaneously transformed into $C_T$ as soon as it enters the ocean. River delivery of the other four nutrients (Fe, N, P, and Si) was calculated from riverine $C_T$ delivery, assuming constant ratios of $C : N : P : Si : Fe = 320 : 16 : 1 : 53.3 : 3.64 \times 10^{-3}$ (Meybeck, 1982). For sediment mobilization, dissolved iron input was parameterized as $2 \, \mu\text{mol Fe m}^{-2} \, \text{d}^{-1}$ for depths shallower than 1100 m following Moore et al. (2004).

## 2.2 Biogeochemical simulations

For initial conditions, we used observational climatologies for temperature and salinity combined from three sources (Barnier et al., 2006): for dissolved oxygen and nutrients (nitrate, $P_T$, and Si$_T$) from the 2001 World Ocean Atlas (Conkright et al., 2002) and for preindustrial $C_T$ and $A_T$ from the observation-based Global Data Analysis Product (GLODAP) (Key et al., 2004). As comparable observational climatologies for DOC and iron are lacking, those variables were initialized from the output of a 3000-year spin-up of an ORCA2 simulation including PISCES. Other tracers have

short recycling times and were thus initialized with globally uniform constants.

For physical boundary conditions, all simulations were forced with the same DRAKKAR forcing set (DFS) constructed originally by Brodeau et al. (2010) and routinely updated. This historical reanalysis-based forcing data set provides surface air temperature and humidity at 2 m, wind fields at 10 m, shortwave and longwave radiation, and the net surface freshwater flux (evaporation minus precipitation). It covers 55 years, including 1958–2001 from version 4.2 and 2002–2012 from version 4.4.

A 50-year spin-up was first made from rest in the ORCA05 NEMO-PISCES model (coupled circulation–biogeochemistry), after initializing the model variables with the abovementioned fields. The resulting simulated physical and biogeochemical fields were then used to initialize the ORCA05 NEMO-PISCES simulations in 1870, and that model was subsequently integrated over 1870–1957. Since no atmospheric reanalysis is available during that period, we simply looped the DRAKKAR Forcing Set. Then, at the beginning of 1958, the ORCA05 simulated fields were interpolated to the ORCA2 and ORCA025 grids, and simulations were continued in each of the three configurations during 1958 to 2012. We refer to these simulations as B1870-ORCA2, B1870-ORCA05, and B1870-ORCA025 (Table 3), where the first letter refers to the type of simulation (B for biogeochemical), the following four numbers refer to the initialization year, and the remainder refers to the resolution used over 1958–2012.

The initialization of the ORCA025 and ORCA2 models in 1958 with interpolated fields from ORCA05 introduces an error into the results from B1870-ORCA2 and B1870-ORCA025. To estimate this branching error in the low-

**Table 3.** Set of simulations.

| Name | Resolution | |
|------|------------|---|
| | Before 1958 | After 1958 |
| **Biogeochemical** | | |
| B1870-ORCA2 | ORCA05 | ORCA2 |
| -ORCA05 | ORCA05 | ORCA05 |
| -ORCA025 | ORCA05 | ORCA025 |
| -ORCA2* | ORCA2 | ORCA2 |
| **Perturbation** | | |
| P1870-ORCA2 | ORCA05 | ORCA2 |
| -ORCA05 | ORCA05 | ORCA05 |
| -ORCA025 | ORCA05 | ORCA025 |
| -ORCA2* | ORCA2 | ORCA2 |
| P1765-ORCA2 | ORCA05 | ORCA2 |
| -ORCA05 | ORCA05 | ORCA05 |
| -ORCA025 | ORCA05 | ORCA025 |
| -ORCA2* | ORCA2 | ORCA2 |

resolution model, we also made a simulation using ORCA2 from 1870 to 2012 (referred to as B1870-ORCA2*) and compared it to B1870-ORCA2 (initialized in 1958 with output from ORCA05). This strategy was not possible for ORCA025 because running ORCA025 with PISCES over 1870–2012 is too costly.

For each member of this "B" class of simulations, we actually made two types of runs: historical and control, both forced with the same reanalysis fields. Those two runs differ only in their atmospheric $CO_2$ concentrations. The control simulations were forced with the preindustrial $CO_2$ concentration of 287 ppm in the atmosphere over the entire period from 1870 to 2012. The historical simulations were forced with yearly averaged historical atmospheric $CO_2$ concentrations reconstructed from ice cores and atmospheric records over 1870 to 2012 starting at the same reference of 287 ppm (Le Quéré et al., 2015). Making both the control and the historical runs for each of the B class of simulations and taking the difference automatically corrects for model drift. That CE7 difference is defined as the anthropogenic component.

## 2.3 $C_{ant}$ perturbation simulations

Because of computational limitations, it was necessary to start the anthropogenic $CO_2$ perturbation of our reference ORCA05-PISCES simulation in 1870 as opposed to the traditional earlier reference of 1765 (Sarmiento et al., 1992), a more realistic approximation of the start of the industrial-era $CO_2$ increase. A similar compromise was adopted for the Coupled Model Intercomparison Project Phase 5 (CMIP5) (Taylor et al., 2012). During the missing 105 years, atmo-

spheric $xCO_2$ increased from 278 to 287 ppm, a 9 ppm difference that seems small relative to today's total perturbation with atmospheric $xCO_2$ now above 400 ppm. However, Bronselaer et al. (2017) estimated that global ocean uptake of $C_{ant}$ in 1995 is actually underestimated by $\sim 30\%$ (29 Pg C) for simulations that reference the natural preindustrial state to 1850 rather than 1765. The cause is partly due to ocean carbon uptake during the missing 1765–1850 period, but mostly it is due to the higher preindustrial reference for atmospheric $xCO_2$ that results in the air–sea flux of $C_{ant}$ being underestimated throughout the entire simulation. Unfortunately, we cannot use the Bronselaer et al. (2017) results to correct our biogeochemical simulations because they do not include the Arctic Ocean in their global data-based assessment. Furthermore, their reference date in the mid 19th century is 20 years earlier than ours.

Instead, to correct for the late starting date of our biogeochemical simulations, we made additional simulations using the more efficient single-tracer perturbation approach (Sarmiento et al., 1992) rather than the full PISCES biogeochemical model (24 tracers). To account for the missing carbon, we added the difference between two perturbation simulations, denoted as "P" rather than B, with one starting in 1765 (P1765) and the other starting in 1870 (P1870). For consistency, we applied the same initialization strategy as for the biogeochemical simulations, i.e., using ORCA05 until the end of 1957 with that output serving as the initial fields for subsequent 1958–2012 simulations in all three configurations. The naming convention for the P class of simulations is like that for the B class (indicated by the first letter). The difference is that in each P simulation there is only one tracer and one run for each (no need for a control and a historical run). However, initializing a set of P simulations in 1765 as well as in 1870 implies twice the number of simulations (Table 3). The difference in $C_{ant}$ between P1765 and P1870 simulations was later added as a late-start correction to the biogeochemical simulations (B1870), for each resolution separately.

The perturbation approach of Sarmiento et al. (1992) avoids the computationally intensive standard $CO_2$ system calculations by accounting for only the perturbation ($C_{ant}$), assuming it is independently of the natural carbon cycle. By focusing only on anthropogenic carbon, this approach exploits a linear relationship between the anthropogenic change in surface-ocean $pCO_2$ (µatm) and its ratio with the corresponding change in surface-ocean dissolved inorganic carbon ($\delta C_T$):

$$\frac{\delta pCO_{2o}}{\delta C_T} = z_0 + z_1 \delta pCO_{2o},$$ (1)

where $\delta pCO_{2o}$ is the perturbation in surface-ocean $pCO_2$ and the coefficients $z_0$ and $z_1$ are each quadratic functions

**Table 4.** Fitted parameters in Eqs. (2) and (3) for the perturbation simulations P1765 and P1870.

| Parameter | P1765 | P1870 |
|---|---|---|
| $a_0$ | 1.7481 | 1.8302 |
| $a_1$ | $-3.2813 \times 10^{-2}$ | $-3.4631 \times 10^{-2}$ |
| $a_2$ | $4.1855 \times 10^{-4}$ | $4.3614 \times 10^{-4}$ |
| $b_0$ | $3.9615 \times 10^{-3}$ | $4.0105 \times 10^{-3}$ |
| $b_1$ | $-7.3733 \times 10^{-5}$ | $-7.3386 \times 10^{-5}$ |
| $b_2$ | $5.4759 \times 10^{-7}$ | $5.1199 \times 10^{-7}$ |

of surface temperature (°C):

$$z_0 = a_0 + a_1 T + a_2 T^2, \qquad (2)$$

$$z_1 = b_0 + b_1 T + b_2 T^2. \qquad (3)$$

In the model, Eq. (1) was rearranged to solve for $\delta p\mathrm{CO}_{2o}$ in terms of $\delta C_T$ (Sarmiento et al., 1992, Eq. 11), as needed to compute the air–sea flux (Sarmiento et al., 1992, Eq. 2) CE8. In the air–sea flux equation, the atmospheric $x\mathrm{CO}_2$ was corrected for humidity and atmospheric pressure to convert to $p\mathrm{CO}_{2\mathrm{atm}}$, which thus varies spatially while atmospheric $x\mathrm{CO}_2$ does not (in the model). The atmospheric $x\mathrm{CO}_2$ history for 1765–1869 is from Meinshausen et al. (2017), while the history for 1870 and beyond is the same as used in the NEMO-PISCES simulations. One set of coefficients was derived for our reference atmospheric $x\mathrm{CO}_2$ in 1765; another set was derived for our reference atmospheric $x\mathrm{CO}_2$ in 1870 (Table 4). The original approach was only updated to use the equilibrium constants recommended for best practices (Dickson et al., 2007) and to cover a perturbation of up to 280 ppm (see Supplement). The relative uncertainty introduced by approximating the perturbation to the ocean $\mathrm{CO}_2$ system equilibria with Eq. (1) remains less than $\pm 0.3\%$ across the global ocean's observed temperature range when $\delta p\mathrm{CO}_2{}^{\mathrm{oc}} < 280$ ppm.

The Arctic $C_{\mathrm{ant}}$ inventory in 2012 simulated by the perturbation approach (P1870-ORCA2*) underestimates that simulated by the full biogeochemical approach (B1870-ORCA2*) by 3 % (Appendix A). The corresponding underestimation by P1765-ORCA2* is expected to be similar. The similar bias of P1765-ORCA2* and P1870-ORCA2* means that the bias in their difference is probably much less than 3 %. The same holds for P1765-ORCA2 vs. P1870-ORCA2. Thus, using their difference to correct for the late start of B1870-ORCA2 is not only practical but also sufficiently accurate for our purposes. In contrast, not correcting B1870-ORCA2 for its late starting date would lead to a 19 % underestimation of its Arctic $C_{\mathrm{ant}}$ inventory for the full industrial era.

In practice, to make the late-start correction, at each grid cell we added the time-varying difference in $C_{\mathrm{ant}}$ between the two perturbation simulations (P1765 − P1870) to the $C_{\mathrm{ant}}$ simulated with B1870 for each resolution separately. From here on, we refer only to these corrected biogeochemical sim-

ulations, denoting them as ORCA2, ORCA05, ORCA025, and ORCA2*.

## 2.4 CFC-12 simulation

CFC-12 is a purely anthropogenic tracer, a sparingly soluble gas whose concentration began to increase in the atmosphere in the early 1930s and part of which has been transferred to the ocean via air–sea gas exchange. Its uptake and redistribution in the ocean has been simulated following OCMIP-2 CE9 protocols (Dutay et al., 2002). The CFC-12 flux ($F_{\mathrm{CFC}}$) at the air–sea interface was calculated as follows:

$$F_{\mathrm{CFC}} = k_w(\alpha_{\mathrm{CFC}} p\mathrm{CFC} - C_s)(1 - I), \qquad (4)$$

where $k_w$ is the gas-transfer velocity (piston velocity) in $\mathrm{m\,s}^{-1}$ (Wanninkhof, 1992), $p\mathrm{CFC}$ is the atmospheric partial pressure of CFC-12 from the reconstructed atmospheric history by Bullister (2015), $C_s$ is the sea surface concentration of CFC-12 ($\mathrm{mol\,m}^{-3}$), $\alpha_{\mathrm{CFC}}$ is the solubility of CFC-12 ($\mathrm{mol\,m}^{-3}\,\mathrm{atm}^{-1}$) from Warner and Weiss (1985), and $I$ is the model's fractional sea-ice cover. Once in the ocean, CFC-12 is an inert tracer that is distributed by advection and diffusion; it has no internal sources and sinks. Many high-precision measurements of CFC-12 are available throughout the ocean, in sharp contrast to $C_{\mathrm{ant}}$ which cannot be measured directly.

As for the other simulations, those for CFC-12 were made using ORCA05 until 1957, at which point those results were interpolated to the ORCA2 and ORCA025 grids. The ORCA05 CFC-12 simulation began in 1932. From 1958 to 2012, CFC-12 was simulated for each resolution separately.

## 2.5 Arctic Ocean

To assess the anthropogenic carbon budget in the Arctic Ocean, we adopt the regional domain defined by Bates and Mathis (2009) delineated in Fig. 1. That domain's lateral boundaries and the volume of water contained within them vary slightly among the three model versions due to their different resolutions and bathymetries (Table 1). The signature of these different volumes is also apparent in the integrated quantity of anthropogenic carbon that is stored in the Arctic in 1958, although the fields for all three models are based on the same 1957 field from the ORCA05 model (Fig. 2).

## 2.6 Transport across boundaries

Transects are defined (Fig. 1) along the four boundaries as consistently as possible for the three resolutions. Water transport across each of the four boundaries is calculated for each model configuration by using monthly average water velocities at each boundary grid cell along a transect multiplied by the corresponding area of the face of the grid cell through which the water flows. For boundaries defined by a row of cells (Fram Strait; Canadian Arctic Archipelago, CAA; and

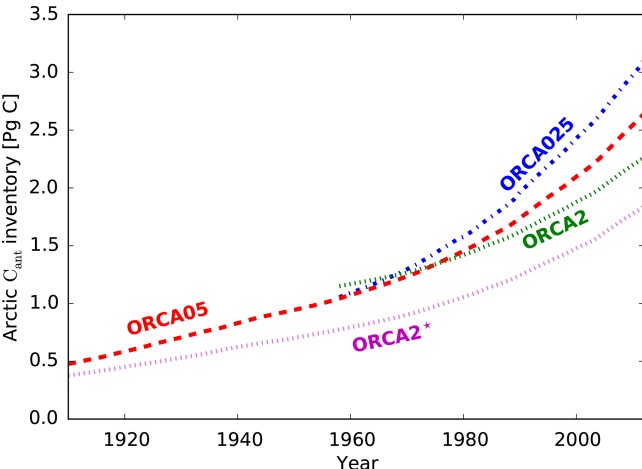

**Figure 2.** Arctic Ocean $C_{ant}$ inventory for ORCA2, ORCA05, ORCA025, and ORCA2*. The discontinuity for ORCA2 in 1958 is due to its larger total volume of water when integrated across the Arctic domain (Table 1).

Bering Strait), the transport is calculated across the northern face of each cell. Conversely, for the jagged boundary of the Barents Sea Opening, transport is calculated at the northern and eastern faces of each cell and the two transports are summed. For each transect, transport across all of its cells is summed to obtain the transect's monthly net transport.

For the $C_{ant}$ transport, we do the same but also multiply the water transport at the boundary between two grid cells with their volume-weighted monthly-average concentration. This multiplication of monthly means introduces an error into the transport calculations owing to neglect of shorter-term variability. To elucidate that error, we sum results from those monthly calculations across all four sections, integrate them over time from 1960 to 2012, and compare that to the net transport of $C_{ant}$ into the Arctic Ocean implied by the inventory change minus the cumulative air–sea flux over the same time period. The inventory of $C_{ant}$ is the total mass of $C_{ant}$ inside the Arctic Ocean at a given time, while the cumulative flux is the time-integrated air–sea flux of anthropogenic $CO_2$ over the Arctic Ocean since the beginning of the simulation. The difference between these two spatially integrated values is the reference value for the net lateral flux into the Arctic Ocean to which is compared the less exact total lateral flux of anthropogenic carbon computed from monthly mean velocity and concentration fields integrated over time. The relative error for transport of $C_{ant}$ across all the separate boundaries introduced by the monthly average calculations is 28 % for ORCA2, 7 % for ORCA05, and 3 % for ORCA025. This error applies neither to the $C_{ant}$ inventory nor to the cumulative $C_{ant}$ air–sea flux or the lateral $C_{ant}$ fluxes, which are all calculated online, i.e., during the simulations.

## 2.7 CFC-12 observational data

Model simulations were evaluated indirectly by comparing simulated to observed CFC-12. We choose CFC-12 to evaluate the model because it is an anthropogenic, passive, conservative, and inert tracer, and in contrast to anthropogenic carbon, it is directly measurable. The CFC-12 atmospheric concentration increased from zero in the 1930s to its peak in the 2000s, since declining as a result of the Montreal protocol. Thus, CFC-12 is a transient tracer similar to anthropogenic carbon but for which there exist extensive direct measurements, all carried out with high precision during the WOCE (World Ocean Circulation Experiment) and CLIVAR (Climate and Ocean – Variability, Predictability and Change) era. Nowadays, ocean models are often evaluated with CFC-11 or CFC-12, especially those destined to be used to assess anthropogenic carbon uptake (Dutay et al., 2002; Orr et al., 2017).

The CFC-12 observations used in this study come from two trans-Arctic cruises: the 1994 Arctic Ocean Section (AOS94) (Jones et al., 2007) and the Beringia 2005 expedition (Anderson et al., 2011) (Fig. 1). AOS94 started on 24 July and was completed on 1 September, during which CFC-12 measurements were made at 39 stations. That section began in the Bering Strait, entered the Canada Basin adjacent to Mendeleev Ridge, continued to the Makarov Basin, and ended at the boundary of the Nansen Basin and the Barents Sea. The Beringia expedition started on 19 August and ended on 25 September 2005. It began off the coast of Alaska, went through the Canada and Makarov basins and crossed the Lomonosov Ridge, and its last CFC-12 station was taken on the Gakkel Ridge. These two cruises were chosen from among others because they cross large parts of the Arctic, including almost all four major basins.

## 2.8 Data-based estimates of anthropogenic carbon

Our simulated $C_{ant}$ was compared to data-based estimates from Tanhua et al. (2009) for the year 2005 and from GLODAPv2 for the year 2002 (Lauvset et al., 2016), both based on the TTD approach.

## 3 Results

### 3.1 Physical evaluation

#### 3.1.1 Lateral water fluxes

The lateral water flux across each of the four Arctic boundaries is a fundamental reference for the simulated physical transport, especially given our goal to construct a budget that includes the lateral transport of passive tracers. Results for lateral water transport in the three model resolutions may be grouped into two classes: coarse resolution and higher resolutions. In ORCA2, water enters the Arctic Ocean from

the Barents Sea and the Bering Strait (2.1 Sv split evenly), with 86 % of that total leaving the Arctic via the Fram Strait and the remaining 14 % flowing out via the CAA (Table 5). Conversely, outflow through the CAA is 7 times larger for ORCA05 and 9 times larger for ORCA025, being fueled by 26 % to 46 % more inflow via the Bering Strait and 110 % to 170 % more inflow via the Barents Sea. Outflow via the Fram Strait is 1.76 Sv in ORCA2, 1.42–1.75 Sv in ORCA05, and 1.46–1.80 Sv in ORCA025, depending on the time period (Table 5).

Relative to the observed CAA outflow of 2.7 Sv (Curry et al., 2014; Straneo and Saucier, 2008), only ORCA05 and ORCA025 simulate similar results. In contrast, ORCA2's simulated CAA outflow is about one-ninth of that observed. Likewise, its inflow via the Barents Sea is half of that observed, while the two higher-resolution simulations have Barents Sea inflows that are 20 % and 40 % larger than observed. Yet for inflow through the Bering Strait, it is ORCA2 that is closest to the observed estimate, overestimating it by 30 %, while ORCA05 and ORCA025 overestimate it by 60 % and 90 %. Thus, too much Pacific water appears to be entering the Arctic Ocean. All resolutions underestimate the central observational estimate for the Fram Strait outflow by ∼ 12 % but still easily fall within the large associated uncertainty range.

Summing up, the net water transport across all four boundaries is not zero. A net Arctic outflow between 0.12 and 0.17 Sv is found for the three model resolutions owing to river inflow and precipitation as well as artifacts caused by using monthly averages. In contrast, when the observed water transport estimates at all four boundaries are summed up, there is a net outflow of 1.9 Sv, more than 10 times larger. This strong net outflow is also much larger than freshwater input from rivers of 0.08 Sv (McClelland et al., 2006) and precipitation of 0.12 Sv (Yang, 1999). It can only be explained by uncertainties in the data-based estimates of water transport, which are at least ±2.7 Sv for the net transport based on the limited uncertainties available for transport across the individual boundaries (Table 5). The excessive central observational estimate for the net outflow might be explained by a data-based estimate for the Barents Sea inflow that is too weak combined with a data-based estimate for the Fram Strait outflow that is too strong, a possibility that is consistent with results from the higher-resolution models ORCA05 and ORCA025.

### 3.1.2 Sea ice

Because sea-ice cover affects the air–sea $CO_2$ flux and hence anthropogenic carbon concentrations in the ocean, we compare the modeled sea-ice cover to that observed by the US National Snow and Ice Data Center (Walsh et al., 2015). Yearly averages of sea-ice extent agree within 2 % between the observations and models. Only in summer are simulated sea-ice concentrations slightly too high (by 0.25–0.5 ×

$10^6$ km$^3$, e.g., 5 %). Despite this overall agreement in integrated sea-ice extent, regionally differences are larger. During winter (Fig. 3), all three model configurations slightly overestimate the sea-ice extent northeast of Iceland and north of the Labrador Sea, while the simulated sea-ice extent in the Barents Sea and the Bering Strait are similar to observations. During summer, the simulated sea-ice extent resembles that observed in the western Arctic particularly near the Pacific, but all model resolutions slightly overestimate sea-ice extent in the Nordic Seas, north of the Barents Sea, the Kara Sea, and the Laptev Sea. This overestimation should reduce air–sea $CO_2$ fluxes locally in these regions. Overall, the close model–data agreement for sea-ice extent in terms of the total amount, its trend and seasonal coverage, as well as regional coverage in winter contrasts with the tendency of the models to overpredict sea-ice cover in summer at the highest latitudes of the eastern Arctic.

### 3.1.3 Atlantic water

In the Arctic Ocean, water temperature is used to help identify water masses, with values above 0 °C typically coming from the Atlantic Ocean (Woodgate, 2013). The observed temperature along the 1994 and 2005 sections (Fig. 4) indicates that Atlantic Water (AW) is found between 200 and 1000 m, penetrating laterally below the strongly stratified Arctic Ocean surface waters. In ORCA025, this AW layer is deeper and more diffuse, lying between 500 and 1500 m and thus leading to a cold bias around 500 m and a warm bias around 1000 m. The Beringia station at the boundary between the Barents Sea and the Nansen Basin indicates that AW lies between 200 m (2.5 °C) and the seafloor at 1000 m (0 °C). Conversely in the same location in ORCA025, model temperatures remain above 1.5 °C throughout the water column. That lower maximum temperature and weaker vertical gradient suggest that when ORCA025's Atlantic water enters the Arctic Ocean through the Barents Sea, it is too diffuse, being well mixed throughout the water column. Weaker maxima in ORCA025's simulated temperature relative to observations are also found further west in the Canada Basin along both sections. There, observed temperatures reach maxima of 1.1 °C, while its CE11 simulated maxima reach only 0.5 °C.

The two lower resolutions represent lateral invasion of AW less successfully than does ORCA025. Both simulations show water with temperatures higher than 0 °C only at the southern end of the Nansen Basin. Vertically, these water masses are situated around 400 m for ORCA2 and between 200 and 1300 m for ORCA05.

### 3.2 CFC-12

Simulated CFC-12 was compared among the three resolutions and with observations, focusing first on basin-scale tendencies based on vertical profiles of the distance-weighted means along the Beringia 2005 section (Fig. 5). This compar-

Biogeosciences, 16, 1–25, 2019

**Table 5.** Lateral transport of water and $C_{ant}$ across Arctic Ocean boundaries with average simulated values calculated for the same time period as observations.

| | Model configuration | | | Observations | Year | Sources |
|---|---|---|---|---|---|---|
| | ORCA2 | ORCA05 | ORCA025 | | | |
| Lateral water transport (Sv) | | | | | | |
| Fram Strait | −1.76 | −1.75 | −1.80 | −2.0 ± 2.7 | 1997–2006 | Schauer et al. (2008) |
| | −1.76 | −1.42 | −1.46 | −1.7 | 1980–2005 | Rudels et al. (2008) |
| Barents Sea | 1.20 | 2.50 | 2.77 | 2.0 | 2003–2005 | Skagseth et al. (2008) |
| | 1.04 | 2.42 | 2.78 | 2.0 | 1997–2007 | Smedsrud et al. (2010) |
| Bering Strait | 1.02 | 1.29 | 1.49 | 0.8 ± 0.2 | 1991–2007 | Woodgate et al. (2010) |
| CAA | −0.29 | −2.00 | −2.59 | −2.7 ± 0.2 | 2004–2013 | Curry et al. (2014) |
| Sum | −0.12 | −0.16 | −0.18 | | | |
| | Model configuration | | | Observations | Year | Sources |
| | ORCA2 | ORCA05 | ORCA025 | | | |
| Lateral $C_{ant}$ fluxes ($Tg\,C\,yr^{-1}$) | | | | | | |
| Fram Strait | −17 | −12 | −8 | −1 ± 17 | 2002 | Jeansson et al. (2011) |
| | −17 | −7 | 5 | −12 | 2012 | Stöven et al. (2016) |
| Barents Sea | 16 | 43 | 50 | 41 ± 8 | 2002 | Jeansson et al. (2011) |
| Bering Strait | 18 | 22 | 27 | 18 | 2000–2010* | Olsen et al. (2015) |
| CAA | −5 | −28 | −36 | −29 | 2000–2010* | Olsen et al. (2015) |
| Sum | 18 | 29 | 38 | 29 | 2000–2010* | Olsen et al. (2015) |

* Observational year or period impossible to identify exactly as $C_{ant}$ and velocity measurements are not from the same year.

ison reveals that among resolutions, simulated CFC-12 concentrations differ most between 400 and 1900 m; conversely, above and below that intermediate zone, simulated average profiles are nearly insensitive to resolution. In that intermediate zone and above, simulated concentrations are also generally lower than observed. The only exception is the top 100 m in the Canada Basin where all resolutions overestimate observed concentrations by 10 %. Between 200 and 400 m, all resolutions underestimate observations by ∼ 50 %. Below 400 m, the ORCA2 CFC-12 concentrations decline quickly to zero by ∼ 1000 m, while the ORCA05 and ORCA025 concentrations continue to increase, both being 15 % greater at 900 m than at 400 m. Below that depth, the ORCA05 concentrations decline, quickly reaching zero at 1350 m, while ORCA025 concentrations remain above $1\,pmol\,kg^{-1}$ until 1400 m. Between 1100 and 1500 m, average CFC-12 concentrations along the Beringia section in ORCA025 are up to ∼ 10 % larger at 1300 m than observed. This overestimation of CFC-12 by ORCA025 reaches up to 40 % in the Canada and Makarov basins. Below 1900 m, the simulated concentrations are essentially zero, while the observations are slightly higher ($0.12\,pmol\,kg^{-1}$). For comparison, the re-

ported detection limit for CFC-12 for the Beringia 2005 expedition is $0.02\,pmol\,kg^{-1}$ (Anderson et al., 2011).

Given the closer overall agreement of the ORCA025 simulated CFC-12 to the observations, let us now focus on its evaluation along the 1994 and 2005 sections (Fig. 6). On the Atlantic end of the Beringia 2005 section, where water enters the Nansen Basin from the Barents Sea, the water column in ORCA025 appears too well mixed, having CFC-12 concentrations that remain above $2.0\,pmol\,kg^{-1}$. Conversely, observed CFC-12 is less uniform, varying from $2.8\,pmol\,kg^{-1}$ at the surface to $1.3\,pmol\,kg^{-1}$ in bottom waters at 1000 m, thereby indicating greater stratification. A similar contrast in stratification was deduced from modeled and observed temperature profiles at the same location (Sect. 3.1.3). On the other side of the Arctic in the Canada Basin, there are observed local chimneys of CFC-12 where concentrations remain at about $2.0\,pmol\,kg^{-1}$ from near the surface down to 1000 m, particularly along the 1994 section. These chimneys suggest localized mixing that is only barely apparent in ORCA025 (Fig. 6). Such localized features are absent at a lower resolution. The CFC-12 inventories were also calculated along the two sections, integrated over depth and distance (Table 7). Depending on the expedition, ORCA025

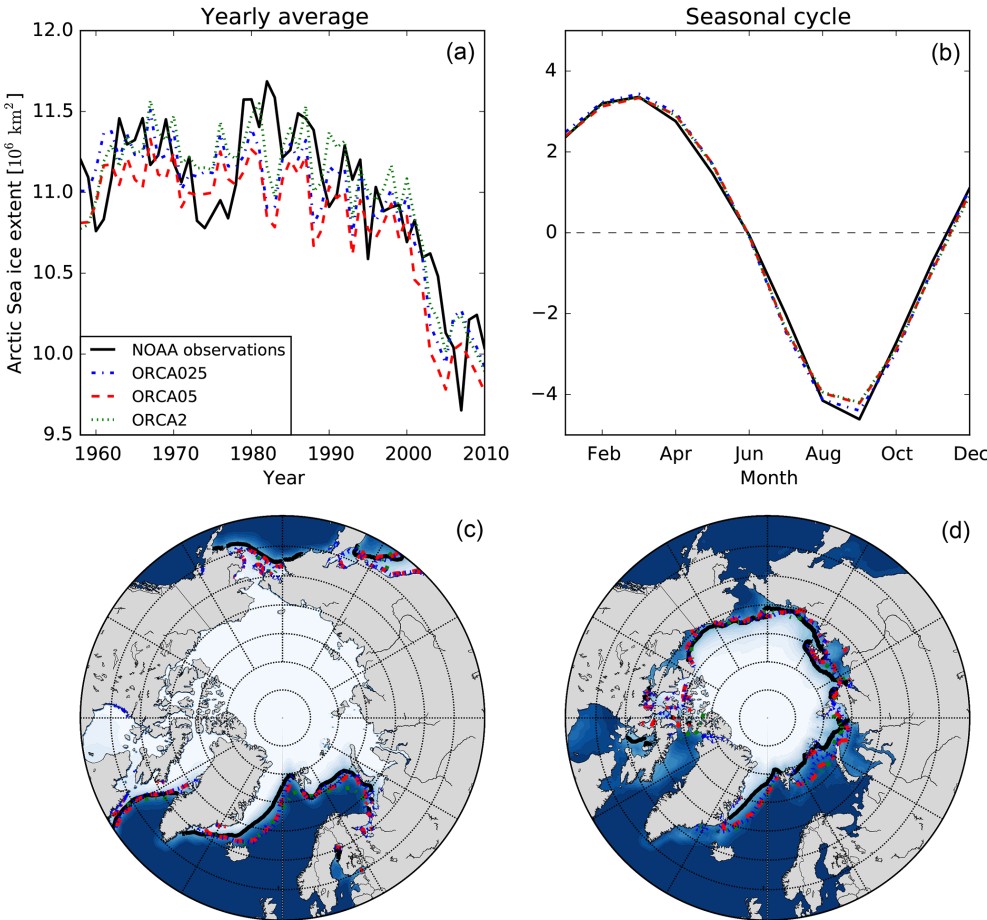

**Figure 3.** Sea-ice extent **(a, b)** TS4 and sea-ice concentration **(c, d)** over the Arctic from 1960 to 2012 comparing microwave-based observations from NOAA CE10 (black) to simulated results from ORCA2 (green dots), ORCA05 (red dashes), and ORCA025 (blue dot–dash). Shown are the yearly averages **(a)**, the average (climatological) seasonal cycle over 1958–2010 **(b)**, and the average sea-ice extent in winter (December, January, February) **(c)** and summer (July, August, September) **(d)**. The lines on the maps show the 50 % sea-ice cover for the three model resolutions and the observations, while the color indicates the observed sea-ice concentration.

underestimates the observed CFC-12 section inventories by 13 %–18 %, ORCA05 by 36 %–38 %, and ORCA2 by 47 %–61 %.

### 3.3 Anthropogenic carbon inventories and concentrations

Simulated global ocean $C_{ant}$ inventories are 152 Pg C in ORCA2, 146 Pg C in ORCA05, and 148 Pg C in ORCA025 in 2008, after accounting for corrections for the earlier starting date of 1765 using our perturbation simulations (P1765–P1870). The correction is similar for each resolution, e.g., 24–25 Pg C in 1995, and is consistent with our biogeochemical model simulation strategy (all three resolutions initialized with the ORCA05 output in 1958). Furthermore, these model-based corrections fall within the $29 \pm 5$ Pg C correction calculated for the same 1765–1995 period with a data-based approach (Bronselaer et al., 2017). For the 1765–2008 period, the data-based global $C_{ant}$ inventory estimate from

(Khatiwala et al., 2009) is $140 \pm 24$ Pg C, the range of which encompasses the results from all three model resolutions.

In the Arctic Ocean, the corrected modeled $C_{ant}$ inventories range from 1.9 to 2.5 Pg C in 2002 and from 2.0 to 2.6 Pg C in 2005, in each case with the low from ORCA2 and the high from ORCA025 (Table 6 and Fig. 2). These simulated basin-wide Arctic Ocean $C_{ant}$ inventories were compared to the TTD-based estimates of anthropogenic carbon from (1) the GLODAPv2 assessment (Lauvset et al., 2016) normalized to the year 2002 and (2) the Tanhua et al. (2009) assessment normalized to 2005. The data-based assessment from GLODAPv2 suggests that 2.9 Pg C of anthropogenic carbon was stored in the Arctic Ocean in 2002, while that from Tanhua et al. (2009) CE13 suggests that 2.5–3.3 Pg C was stored there in 2005. In 2002, the upper limit of the modeled $C_{ant}$ inventory range remains 0.4 Pg C lower than the GLODAPv2 data-based estimate, but the ORCA025 result in 2005 falls within the data-based uncertainty range of Tanhua

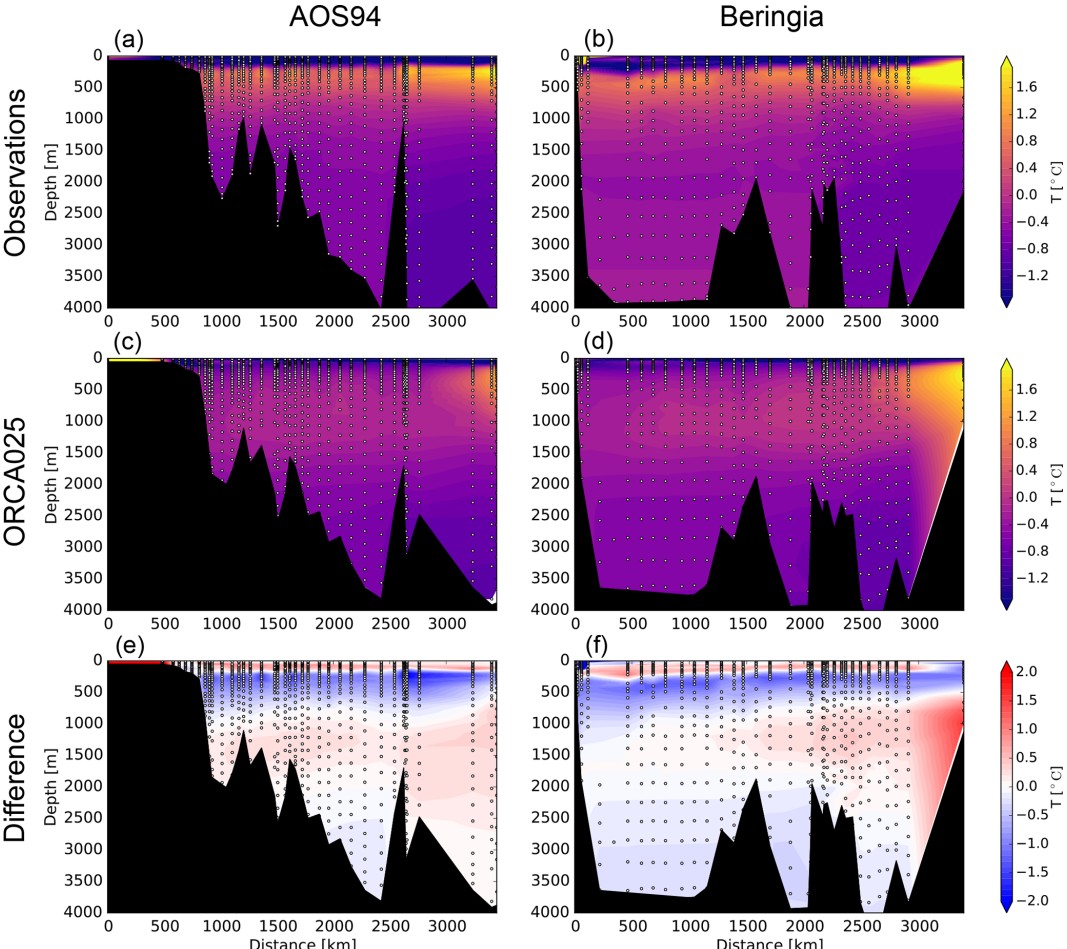

**Figure 4.** Temperature along the 1994 Arctic Ocean Section (AOS94) cruise **(a, c, e)** and the Beringia-HOTRAX CE12 2005 section **(b, d, f)**, both trans-Arctic transects (Fig. 1). The observations **(a, b)** are compared to simulated results from ORCA025 averaged over the summer of the respective year **(c, d)**. The model–data difference is shown at the bottom.

et al. (2009). As for the global estimates, the modeled Arctic Ocean $C_{ant}$ inventories include corrections for the late starting date of the biogeochemical simulations. This correction is 0.4 Pg C in 2005 for each of the three resolutions (Table 6).

The differences in basin-wide inventory estimates were further studied by comparing vertical profiles of $C_{ant}$ from the models to those from the GLODAPv2 data-based estimates (Fig. 7). Surface concentrations in ORCA05 and ORCA025 are up to $\sim 35\%$ larger ($+12\,\mu\mathrm{mol\,kg^{-1}}$) than the data-based estimate, whereas the ORCA2 concentration is $\sim 22\%$ larger ($+7\,\mu\mathrm{mol\,kg^{-1}}$). By 150 m, the simulated concentrations in all resolutions have dropped below the data-based estimates and remain so, except for ORCA025, down to the ocean bottom. Data–model differences are largest at 400 m, with all resolutions underestimating data-based $C_{ant}$ estimates by up to $\sim 28\%$ ($9\,\mu\mathrm{mol\,kg^{-1}}$). Below that depth, results from the three resolutions differ more. The $C_{ant}$ concentration in ORCA2 decreases monotonically reaching $11\,\mu\mathrm{mol\,kg^{-1}}$ at 1000 m and essentially

zero concentration by 2300 m. The vertical penetration of $C_{ant}$ in ORCA2* (the simulation without branching from ORCA05 in 1958) is shallower, reaching zero concentration by 1400 m. In ORCA05, $C_{ant}$ concentrations decrease slowly to $19\,\mu\mathrm{mol\,kg^{-1}}$ at 1000 m, below which they decline rapidly, reaching zero at 2300 m. Only in ORCA025 do $C_{ant}$ concentrations increase again below 400 m, reaching a local maximum at 900 m, an increase that causes the ORCA025 results to exceed data-based estimates by up to $2\,\mu\mathrm{mol\,kg^{-1}}$ ($\sim 11\%$) at 1100 m. A similar maximum and excess are also seen in the CFC-12 profile for ORCA025 as is the minimum around 400 m (Fig. 5). Below 1500 m, the ORCA025 $C_{ant}$ concentrations decline quickly, essentially reaching zero at 2300 m. Conversely, data-based $C_{ant}$ concentrations remain roughly constant at $6\,\mu\mathrm{mol\,kg^{-1}}$ down to the seafloor. Thus, the largest vertically integrated differences between ORCA025 and data-based estimates are found in the deep Arctic Ocean below 1600 m.

**Table 6.** Total inventory, its change during 1960–2012, the cumulative air–sea flux, and the lateral flux of $C_{ant}$ in PgC.

| | Model configuration | | |
|---|---|---|---|
| | ORCA2 | ORCA05 | ORCA025 |
| $C_{ant}$ inventory[a] | | | |
| $C_{ant}$ in 2002[b] | 1.90 (1.47) | 2.25 (1.81) | 2.49 (2.06) |
| $C_{ant}$ in 2005[c] | 1.99 (1.56) | 2.37 (1.96) | 2.64 (2.21) |
| Inventory change (1960–2012) | | | |
| Total Arctic | 1.08 | 1.55 | 1.98 |
| Nansen Basin | 0.14 | 0.33 | 0.30 |
| Amundsen Basin | 0.13 | 0.28 | 0.34 |
| Makarov Basin | 0.15 | 0.21 | 0.33 |
| Canada Basin | 0.31 | 0.36 | 0.61 |
| Cumulative fluxes (1960–2012) | | | |
| Air–sea flux | 0.29 | 0.43 | 0.48 |
| Lateral flux of $C_{ant}^d$ | 0.79 | 1.13 | 1.50 |
| Fram Strait | −0.74 | −0.40 | −0.06 |
| Barents Sea | 0.79 | 1.75 | 1.98 |
| Bering Strait | 0.74 | 0.89 | 1.03 |
| CAA | −0.22 | −1.20 | −1.50 |
| Summed lateral flux | 0.57 | 1.05 | 1.45 |

[a] Numbers in parentheses show the uncorrected value (starting date 1870). [b] Data-based inventory in 2002: 2.95 PgC (GLODAPv2). [c] Data-based inventory in 2005: 3.03 PgC (2.5–3.3) (Tanhua et al., 2009). [d] Computed as inventory change minus cumulative air–sea flux.

**Table 7.** Along-section CFC-12 inventories ($\mu$mol m$^{-1}$) integrated over depth and distance along the AOS94 and Beringia 2005 sections vs. colocated results in ORCA2, ORCA05, and ORCA025.

| | AOS94 | Beringia 2005 |
|---|---|---|
| Observation | 5.5 | 9.4 |
| ORCA2 | 2.9 | 3.7 |
| ORCA05 | 3.5 | 5.8 |
| ORCA025 | 4.8 | 7.7 |

## 3.4 Anthropogenic carbon budget

For the budget of $C_{ant}$, we focused on the final decades over which the model resolutions differed (Tables 5 and 6). During 1960 to 2012, the $C_{ant}$ inventory in ORCA025 increased by 1.98 PgC, 80 % of which is stored in the four major Arctic Ocean basins: the Nansen Basin (0.30 PgC), the Amundsen Basin (0.34 PgC), the Makarov Basin (0.33 PgC), and the Canada Basin (0.61 PgC). Although the Canada Basin $C_{ant}$ inventory increased most, its volume is larger so that its average $C_{ant}$ concentration increased less than in the other basins (Fig. 7). Of the total inventory stored in the Arctic Ocean during that time, only about one-fourth (0.48 PgC) entered the Arctic Ocean via air–sea flux, most of which was transferred from the atmosphere through the surface of the Barents Sea (Fig. 8). The remaining 75 % (1.50 PgC) en-

tered the Arctic Ocean via lateral transport. This net lateral influx is the sum of the fluxes (1) from the Atlantic to the Barents Sea (1.98 PgC), (2) from the Pacific through the Bering Strait (1.03 PgC), (3) to the Atlantic via the Fram Strait (−0.06 PgC), and (4) to the Atlantic via the CAA (−1.50 PgC). Summed up, the net lateral inflow of carbon across the four boundaries is 1.45 PgC. This lateral flux computed from monthly mean $C_{ant}$ concentrations and flow fields is 0.05 PgC ($\sim$ 3 %) smaller than the lateral flux computed from the change in inventory minus the cumulative air–sea flux (Fig. 8). Within the Arctic, coastal regions typically exhibit net lateral losses, while the deep basins exhibit net lateral gain. The largest lateral loss occurs in the Barents Sea, where the cumulative air–sea flux of $C_{ant}$ is also largest.

The budget of $C_{ant}$ changes notably with resolution. Higher resolution results in more simulated $C_{ant}$ being stored in the Arctic region, with increases in both the cumulative air–sea flux and lateral transport. The $C_{ant}$ inventory change from 1960 to 2012 nearly doubles with the resolution increase between ORCA2 and ORCA025 (from 1.08 to 1.98 PgC). Of that additional $C_{ant}$, 93 % is found between 300 and 2200 m, with the maximum being located at 1140 m. The remaining 7 % is located in the upper 300 m (Fig. 7). Resolution also affects the regional partitioning of $C_{ant}$ (Figs. 7 and 8). When refining resolution from ORCA2 to ORCA05, the Arctic Ocean $C_{ant}$ inventory increases by 0.47 PgC, 72 % of which occurs in the Eurasian basins: the

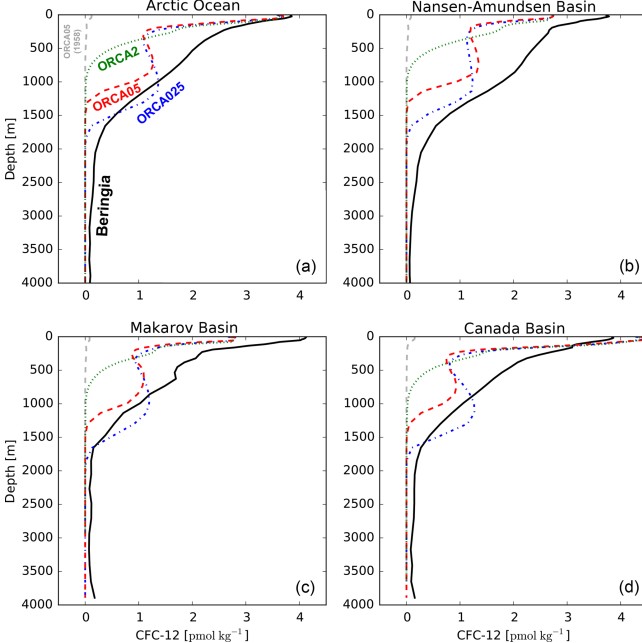

**Figure 5.** Profiles of observed CFC-12 (black solid) and simulated CFC-12 in ORCA2 (green dots), ORCA05 (red dashes), and ORCA025 (blue dot–dash) along the Beringia 2005 section. Shown are distance-weighted means across that entire section **(a)** as well as over that section covering the Nansen and Amundsen basins **(b)**, the Makarov Basin **(c)**, and the Canada Basin **(d)**. Shown in light grey is the vertical profile in 1958, the branching point for the three resolutions.

Nansen Basin (0.19 Pg C) and Amundsen Basin (0.15 Pg C). Another 23 % of that increase occurs in the Amerasian basins: the Makarov Basin (0.06 Pg C) and Canada Basin (0.05 Pg C). Coastal regions account for only 5 % of the total inventory increase. In contrast, the subsequent resolution enhancement between ORCA05 and ORCA025 results in little increase in inventory in the Eurasian basins (0.03 Pg C) but much more in the Amerasian basins (0.37 Pg C).

As resolution is refined between ORCA2 and ORCA025, the Arctic $C_{ant}$ inventory increases as a result of a 66 % increase in the air–sea flux (+0.19 Pg C) and a 90 % increase in the lateral flux (+0.71 Pg C). Thus, the relative contribution of the lateral flux increases from 73 % to 76 %. Changing model resolution also affects the pathways by which $C_{ant}$ enters the Arctic Ocean (Table 5). The most prominent change occurs in the CAA. From ORCA2 to ORCA025, the net outflow of $C_{ant}$ through the CAA increases 7-fold (from −0.22 to −1.50 Pg C). Other notable changes include (1) the net outflow through the Fram Strait declining 12-fold from -0.74 to −0.06 Pg C, (2) the inflow through the Barents Sea increasing by 150 % (from 0.79 to 1.98 Pg C), and (3) the inflow of $C_{ant}$ through the Bering Strait increasing by 39 % (from 0.74 to 1.03 Pg C).

## 4 Discussion

### 4.1 CFC-12

The simulated CFC-12 in ORCA025 underestimates observed concentrations between 100 and 1100 m, slightly overestimates them on average between 1100 and 1500 m, and again underestimates the low observed concentrations below 1500 m. The temperature sections suggest that excess simulated CFC-12 between 1100 and 1500 m is due to a vertical displacement of inflowing Atlantic water, which descends too deeply into the Arctic (Fig. 4). Such vertical displacement would indeed reduce simulated CFC-12 concentrations above 1000 m and enhance them between 1100 and 1500 m. Yet the underestimation of integrated CFC-12 mass above 1100 m is larger than the overestimation below 1100 m. Thus, vertical displacement of Atlantic water cannot provide a full explanation. Simulated CFC-12 concentrations above 1100 m could also be too low because ventilation of subsurface waters is too weak, a hypothesis that is consistent with the simulated vertical gradients in both temperature and CFC-12 that are too strong between 100 and 1100 m.

### 4.2 Anthropogenic carbon

Vertical profiles of $C_{ant}$ and CFC-12 are similar. Above 1000 m, ORCA025 underestimates data-based estimates of $C_{ant}$ as well as observed CFC-12 owing to weak ventilation in the model. Between 1000 and 1500 m, simulated $C_{ant}$ and CFC-12 in ORCA025 exhibit local maxima, which make them on average slightly higher than data-based and observed concentrations. These local maxima can be explained by the simulated Atlantic water masses, rich in both tracers, being too deep. However, the slight overestimation between 1000 and 1500 m is much smaller than the underestimation between 200 and 1000 m. Below 2000 m, simulated $C_{ant}$ largely underestimates data-based estimates. The low simulated $C_{ant}$ stems from too little deep-water formation in the model as indicated by the absence of simulated CFC-12 below 2000 m, an absence that contrasts with the observed CFC-12 concentrations that remain detectable all the way down to the ocean floor (Fig. 5).

A second reason for the low simulated Arctic $C_{ant}$ inventory in ORCA025 is that it was initialized with ORCA05 results in 1958. Had ORCA025 been initialized in 1765, which was not computationally feasible, its simulated inventory would be larger, given that both $C_{ant}$ and CFC-12 storage for ORCA025 are larger than those for ORCA05 over 1958–2012. That hypothesis is consistent with our finding that ORCA2* (complete simulation at 2° without branching from the 0.5° configuration) takes up less $C_{ant}$ than does ORCA2 (0.5° until 1958 then 2° afterwards), which is in line with ORCA2 taking up less $C_{ant}$ and CFC-12 than ORCA05 (Figs. 2, 5, and 7). The initialization of ORCA2 with ORCA05 output in 1958 mainly affects $C_{ant}$ storage be-

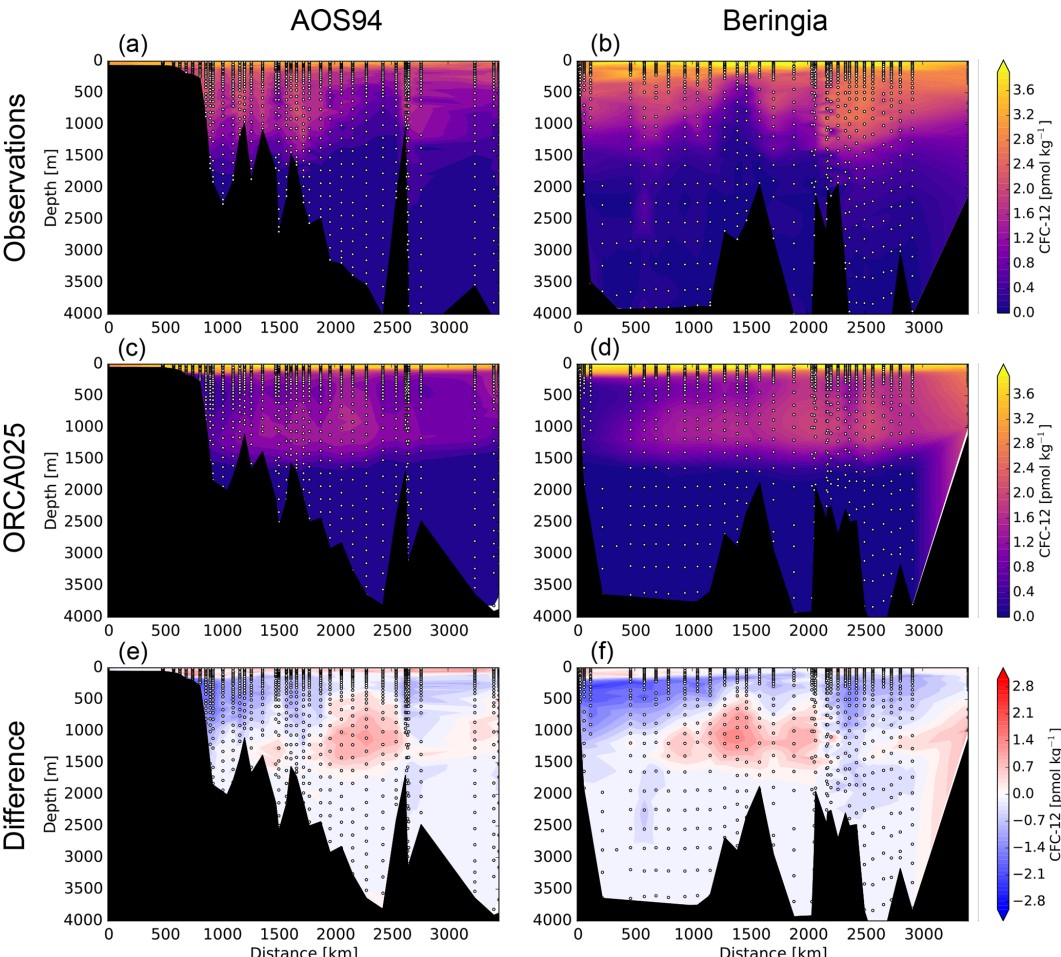

**Figure 6.** CFC-12 concentrations along the AOS94 section **(a, c, e)** and the Beringia section **(b, d, f)** for the observations **(a, b)**, the simulated summer means in ORCA025 **(c, d)**, and the model–data difference **(e, f)**.

tween 1000 and 2000 m, the same depth range over which differences in simulated CFC-12 concentrations are largest between ORCA2 and ORCA05. Nevertheless, that 1958 initialization has little effect on subsequent changes in $C_{ant}$ storage, cumulative lateral flux, and air–sea flux (Fig. 9). Rather it is the changes before 1958 that dominate the difference between ORCA2 and ORCA2*.

All our model configurations underestimate $C_{ant}$ concentrations in the deep waters of the Arctic Ocean based on the CFC-12 model evaluation. The same conclusion is drawn from comparing simulated to data-based estimates of $C_{ant}$. However, results from different data-based approaches to estimate $C_{ant}$ can differ substantially in the deep ocean (e.g., Vázquez-Rodríguez et al., 2009). Furthermore, the TTD approach typically produces the highest values in deep waters due to its assumption of constant air–sea disequilibrium (Khatiwala et al., 2013). Hence applying other data-based approaches to assess the Arctic Ocean inventory of the $C_{ant}$ inventory would eventually help to further constrain uncertainties.

### 4.3 Lateral flux

In our model, about three-fourths of the net total mass of $C_{ant}$ that accumulates in the Arctic Ocean enters laterally from the Atlantic and Pacific oceans, independent of model resolution. Our simulated lateral fluxes of $C_{ant}$ in ORCA025 were compared to data-based estimates from studies that multiply data-based $C_{ant}$ concentrations (TTD estimates) along the Arctic boundaries by corresponding observation-based estimates of water transport.

The simulated lateral transport of $C_{ant}$ in ORCA025 generally agrees with data-based estimates within their large uncertainties. These uncertainties result from uncertainties in data-based estimates of $C_{ant}$ and from uncertainties in observational constraints on water flow, which also varies interannually (Jeansson et al., 2011). For the Fram Strait, Jeansson et al. (2011) estimated a net $C_{ant}$ outflux (from the Arctic) of $1 \pm 17\,\mathrm{Tg\,C\,yr^{-1}}$ in 2002, while for 2012 Stöven et al. (2016) estimate an outflux of $12\,\mathrm{Tg\,C\,yr^{-1}}$ without indicating uncertainties. For the same years, ORCA025 simulates a net

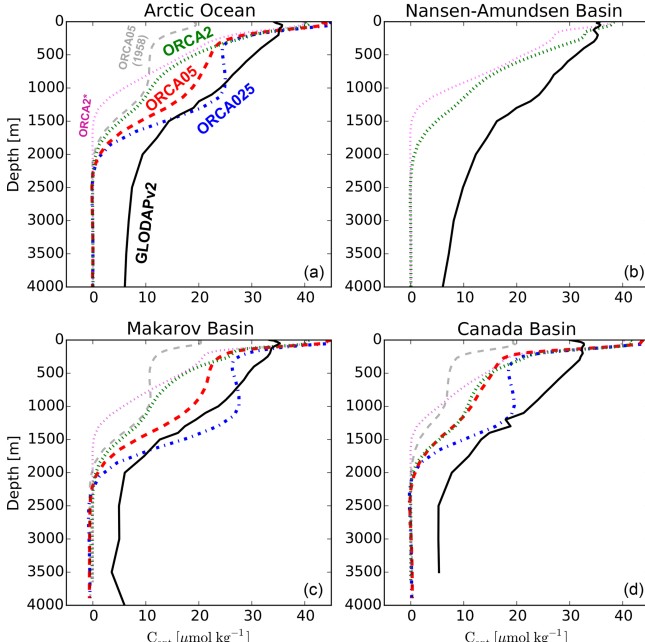

**Figure 7.** Area-weighted basin-wide average vertical profiles of $C_{ant}$ concentration in 2002 for GLODAPv2 data-based estimates (black solid), ORCA2 (green dots), ORCA05 (red dashes), and ORCA025 (blue dot–dash) over the entire Arctic **(a)** as well as over the Nansen and Amundsen basins **(b)**, the Makarov Basin **(c)**, and the Canada Basin **(d)**. Ocean corrected for the starting year by the perturbation approach simulations. Shown are the vertical profile in 1958 (dashed, light grey), the branching point for the three resolutions, and results from ORCA2* for 2002 (magenta dots).

outflux of $8\,\mathrm{Tg\,C\,yr^{-1}}$ in 2002 but a net influx (to the Arctic) of $5\,\mathrm{Tg\,C\,yr^{-1}}$ in 2012. Both model and data-based estimates vary greatly between 2002 and 2012. Across the Barents Sea Opening, there is a consistent net influx from the Atlantic to the Arctic Ocean, i.e., $41\pm8\,\mathrm{Tg\,C\,yr^{-1}}$ in 2002 for the data-based estimate (Jeansson et al., 2011) and $50\,\mathrm{Tg\,C\,yr^{-1}}$ for ORCA025 in the same year.

More recently, Olsen et al. (2015) added data-based estimates of lateral fluxes of $C_{ant}$ across the two other major Arctic Ocean boundaries, completing the set of four boundaries that define the perimeter. They estimate a $C_{ant}$ influx of $\sim18\,\mathrm{Tg\,C\,yr^{-1}}$ from the Pacific through the Bering Strait and a $C_{ant}$ outflux through the CAA of $\sim29\,\mathrm{Tg\,C\,yr^{-1}}$, both for the 2000s. For the same time period, ORCA025 simulates $50\%$ more inflow through the Bering Strait ($\sim27\,\mathrm{Tg\,C\,yr^{-1}}$) and $24\%$ more outflow through the CAA ($\sim36\,\mathrm{Tg\,C\,yr^{-1}}$). The larger Bering Strait $C_{ant}$ influx in ORCA025 is consistent with its overestimated Bering Strait water inflow (Table 5, Sect. 3.1.1). Integrating over all four lateral boundaries, Olsen et al. (2015) found a total net $C_{ant}$ influx of $\sim29\,\mathrm{Tg\,C\,yr^{-1}}$, which is $24\%$ less than that simulated in ORCA025 averaged over 2000–2010 ($\sim38\,\mathrm{Tg\,C\,yr^{-1}}$). Olsen et al. (2015) did not provide uncertainties, but the

uncertainty of their net lateral flux estimate is at least $\pm18\,\mathrm{Tg\,C\,yr^{-1}}$ based on the data-based transport estimates at the two other Arctic boundary sections where uncertainties are available (Table 5).

Weighing in at about one-quarter of the lateral flux is the simulated air–sea flux of $C_{ant}$ in ORCA025 of $10\,\mathrm{Tg\,C\,yr^{-1}}$ when both are averaged over 2000–2010. That simulated estimate is only about $40\%$ of the data-based estimate of $26\,\mathrm{Tg\,C\,yr^{-1}}$ from Olsen et al. (2015). Although no uncertainty is provided with that data-based air–sea flux estimate, it too must be at least $\pm18\,\mathrm{Tg\,C\,yr^{-1}}$ given that it is calculated as the difference between the data-based storage estimate (Tanhua et al., 2009) and the Olsen et al. (2015) data-based net lateral flux. The simulated air–sea flux of $C_{ant}$ falls within that assigned uncertainty range for the data-based estimate. In any case, both the model and data-based estimates suggest that the air–sea flux of $C_{ant}$ is not the dominant contributor to the anthropogenic carbon budget of the Arctic Ocean, respectively representing $21\%$ and $47\%$ of the total $C_{ant}$ input averaged over 2000–2010. For both, the lateral flux dominates.

### 4.4 Model resolution

Basin inventories of simulated anthropogenic carbon differ between model configurations because of how resolution affects their volume, bathymetry, circulation patterns, and source waters. Much of the water in the Nansen and Amundsen basins has entered laterally from the Atlantic Ocean through the Fram Strait and the Barents Sea (Jones et al., 1995). Water inflow through the Barents Sea increases by $150\%$ when moving from ORCA2 to ORCA05 but only by $20\%$ more between ORCA05 and ORCA025. Water inflow in those two higher-resolution models is also closer to observational estimates. Along with the increase in water inflow, higher resolution also increases the lateral influx of $C_{ant}$. Yet despite this increase in the $C_{ant}$ lateral influx, the air–sea $C_{ant}$ flux into the Arctic Ocean also increases with resolution. This finding can be explained by two mechanisms: (1) higher resolution increases the influx of $C_{ant}$ through the Fram Strait, which mainly occurs in subsurface currents and thus does not greatly affect surface $C_{ant}$ concentrations nor air–sea exchanges of $C_{ant}$, and (2) higher resolution enhances deep-water formation, mainly in the Barents Sea, which reduces surface $C_{ant}$ and thus enhances the air-to-sea flux of $C_{ant}$. Although the air–sea flux of $C_{ant}$ increases slightly, the larger lateral water fluxes in ORCA05 and ORCA025 mainly explain their higher $C_{ant}$ concentrations in the Nansen and Amundsen basins. Some of this inflowing water continues to flow further along the slope, across the Lomonosov Ridge into the Makarov Basin, and then across the Mendeleev Ridge into the Canada Basin. Yet how well models simulate that flow path depends on lateral resolution. Between ORCA2 and ORCA05, basin $C_{ant}$ inventories increase by $16\%$ in the Canada Basin ($+0.05\,\mathrm{Pg\,C}$) and by $40\%$ the

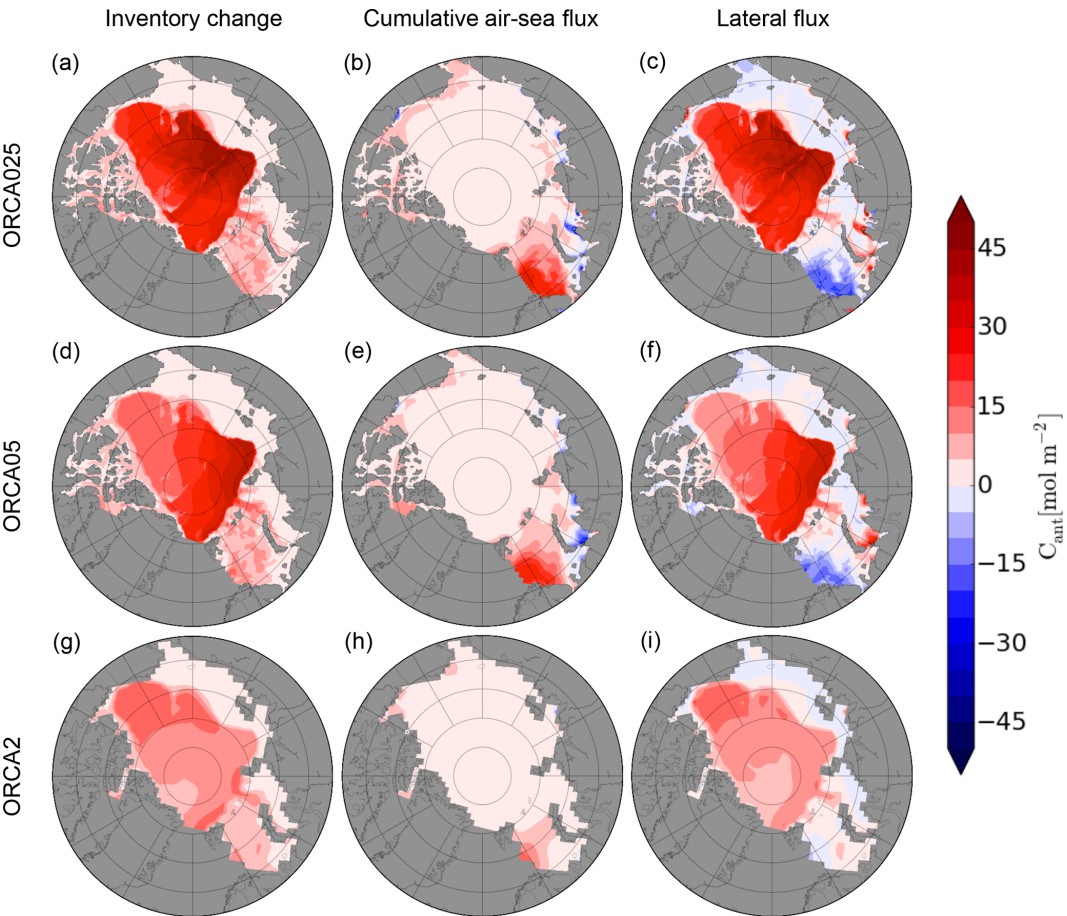

**Figure 8.** Inventory change **(a, d, g)**, cumulative air–sea flux **(b, e, h)**, and the lateral flux calculated as the inventory change minus the cumulative air–sea flux **(c, f, i)** of $C_{ant}$ over 1960–2012 for ORCA025 **(a–c)**, ORCA05 **(d–f)**, and ORCA2 **(g–i)**.

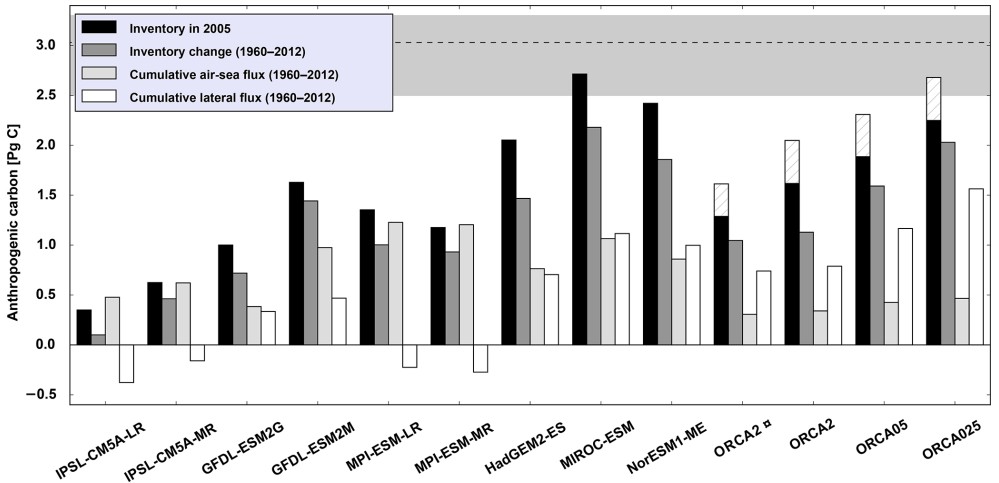

**Figure 9.** Comparison of results for the Arctic Ocean from ORCA2, ORCA05, ORCA025, and ORCA2* and the nine earth system models that participated in CMIP5. Shown are the $C_{ant}$ inventory in 2005 (black), the inventory change of $C_{ant}$ (dark grey) between 1960 and 2012, the corresponding cumulative air–sea flux of $C_{ant}$ (light grey), and the cumulative lateral flux of $C_{ant}$ (white). Also indicated are the data-based estimate from Tanhua et al. (2009) (dashed black line) along with its associated uncertainty range (grey background). The inventory correction for the late starting date for our forced simulations is indicated as striped bars.

Makarov Basin (+0.06 Pg C). But between ORCA05 and ORCA025, increases are 2 to 5 times greater: +0.25 Pg C in the Canada Basin and +0.12 Pg C in the Makarov Basin (Sect. 3.4). The change from ORCA2 to ORCA05 mainly seems to improve lateral exchanges with adjacent oceans, while the change from ORCA05 to ORCA025 improves interior Arctic Ocean circulation.

As the increase from ORCA05 to ORCA025 stems from a finer, more realistic representation of lateral transport within the Arctic, it would appear that eddying ocean models may be needed to adequately simulate the interior circulation in terms of its effect on $C_{ant}$ storage in the Arctic Ocean. In the Canada Basin, such lateral inflow may not be the only source of $C_{ant}$. Another major source appears to come from density flows along the continental slope, driven by brine rejection from sea-ice formation over the continental shelves (Jones et al., 1995). A signature of this source in the observed sections may be the chimneys of constant CFC-12 concentration from the surface to about 1000 m in the Canada Basin, features for which only ORCA025 exhibits any such indication, albeit faint. To adequately model lateral exchanges of $C_{ant}$ in the Arctic Ocean, at least a resolution comparable to that used in ORCA05 may be needed, while resolutions comparable to that in ORCA025 or above may well be required to begin to capture the effects from density flows along the slope. As a consequence of the deficient representation of these density flows, we would expect to see an increase in $C_{ant}$ when using even higher resolution.

Improved modeled circulation from higher model resolution has also been shown to be critical to improving simulated anthropogenic tracers in the Southern Ocean (Lachkar et al., 2007) and simulated oxygen concentrations in the tropical Atlantic (Duteil et al., 2014).

### 4.5   CMIP5 comparison

For a wider perspective, we compared results from our forced NEMO-PISCES simulations to those from nine ocean biogeochemical models that were coupled within different earth system modeling frameworks as part of CMIP5 (Fig. 9). When the CMIP5 models are compared to the data-based estimate of the $C_{ant}$ inventory, only the MIROC-ESM[CE14] with its inventory of 2.7 Pg C falls within the data-based uncertainty estimate (2.5 to 3.3 Pg C in 2005). The next closest CMIP5 models are NorESM1-ME[CE15] and HadGEM2-ES[CE16], which fall below the lower limit of the data-based range by 0.1 and 0.5 Pg C. Then come the MPI-ESM[CE17] and GFDL-ESM[CE18] models with their $C_{ant}$ inventories in 2005 that are 0.9 to 1.5 Pg C lower than the lower limit. The lowest CMIP5 estimates are from both versions of the IPSL[CE19] model, whose inventories reach only ∼ 20 % of the data-based range. Adjusting all the CMIP5-model Arctic inventories upward by ∼ 0.4 Pg C to account for their late start date in 1850, as we did for our three simulations, would place two of them (MIROC-ESM and NorESM1-ME)

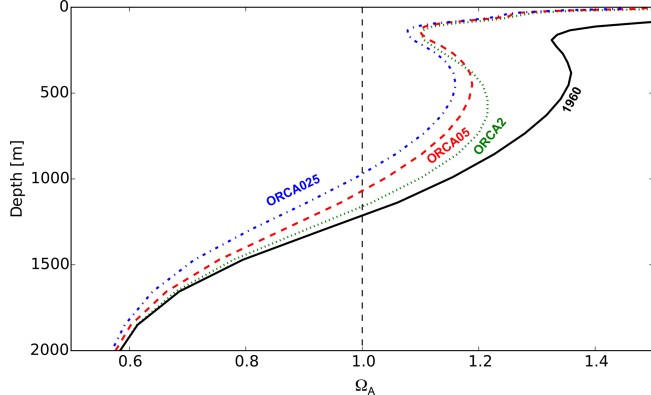

**Figure 10.** Profiles of $\Omega_A$ for ORCA05 in 1960 (black solid) as well as ORCA2 (green dots), ORCA05 (red dashes), and ORCA025 (blue dot–dash) in 2012. The vertical black dashed line indicates the chemical threshold where $\Omega_A = 1$. The point where that vertical line intersects the other curves indicates the depth of the ASH.

above the lower boundary of the data-based uncertainty estimate and another (HadGEM2-ES) just 0.1 Pg C below this lower boundary. Lateral fluxes over 1958–2012 also vary between CMIP5 models from an outflow of 0.3 Pg C in the IPSL-CM5A-LR[CE20] model to an inflow of 1.1 Pg C in the MIROC-ESM model. Only the first three CMIP5 models mentioned above exhibit large net inflow of $C_{ant}$ into the Arctic Basin (between 0.7 and 1.1 Pg C during 1960–2012), a condition that appears necessary to allow a model to approach the estimated data-based inventory range. Indeed, the six other CMIP5 models have lower lateral fluxes (−0.5 to 0.5 Pg C) and simulate low $C_{ant}$ storage in 2005.

What is perhaps most surprising are the large differences between our forced ORCA2* model and the IPSL-CM5A-LR and IPSL-CM5A-MR[CE21] ESMs. All three of those models use the same coarse-resolution ocean model, although both ESMs rely on an earlier version with a different vertical resolution (31 instead of 46 vertical levels). The contrast in vertical resolution may explain part of the large difference in inventories (1.3 Pg C for our forced version that is not corrected for the late starting date vs. 0.3–0.6 Pg C for the two coupled versions) but the forcing could also play a role. Lateral resolution is not the only factor when aiming to provide realistic simulations of $C_{ant}$ storage and lateral transport in the Arctic. Sensitivity studies testing other potentially critical factors are merited.

### 4.6   Effect on aragonite saturation state

Given that simulated $C_{ant}$ is affected by lateral model resolution, so must be simulated ocean acidification. The aragonite saturation state ($\Omega_A$) was computed for each resolution from the historical run's $C_T$, $A_T$, $T$, $S$, $P_T$, and $Si_T$, after correcting $C_T$ and $A_T$ for drift based on the control run. The higher concentrations of $C_{ant}$ in the ORCA05 and ORCA025 sim-

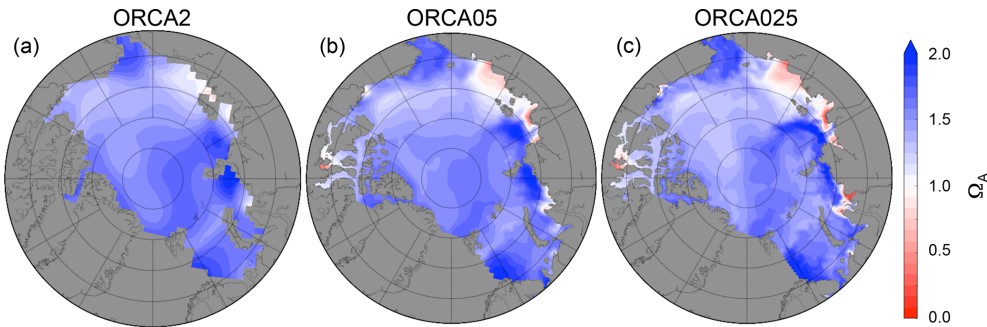

**Figure 11.** Surface $\Omega_A$ for ORCA2, ORCA05, and ORCA025 **(a–c)** in August 2012.

ulations reduces $\Omega_A$ between 1960 and 2012 by more than twice as much as found with ORCA2 during the same period (Fig. 10). These differences translate into different rates of shoaling for the aragonite saturation horizon (ASH), i.e., the depth where $\Omega_A = 1$. During 1960–2012, the ASH shoals by $\sim 50\,\mathrm{m}$ in ORCA2, while it shoals by $\sim 150\,\mathrm{m}$ in ORCA05 and $\sim 210\,\mathrm{m}$ in ORCA025. Thus, model resolution also affects the time at which waters become undersaturated with respect to aragonite with higher resolution producing greater shoaling.

Although basin-wide mean surface $\Omega_A$ does not differ among resolutions, there are regional differences such as over the Siberian Shelf (Fig. 11). The minimum $\Omega_A$ in that region reaches 0.9 in ORCA2, while it drops to 0.3 in ORCA05 and 0.1 in ORCA025. That lower value in ORCA025 is more like that observed, e.g., down to 0.01 in the Laptev Sea (Semiletov et al., 2016). As these lows in $\Omega_A$ are extremely local, they cannot be expected to be captured in coarse-resolution models such as ORCA2. Higher-resolution models are needed in the Arctic to assess local extremes not only in terms of ocean acidification but also other biogeochemical variables.

## 5 Conclusions

Global-ocean biogeochemical model simulations typically have coarse resolution and tend to underestimate the mass of $C_{ant}$ stored in the Arctic Ocean. Our sensitivity tests suggest that more realistic results are offered by higher-resolution model configurations that begin to explicitly resolve ocean eddies. Our high-resolution model simulates an Arctic $C_{ant}$ inventory of 2.6 Pg C in 2005, falling within the uncertainty from the data-based estimates (2.5–3.3 Pg C). That model estimate should be considered a lower bound because it generally underestimates CFC-12 concentrations. Thus, it essentially confirms the lower bound from the data-based estimates, which are based on CFC-12-derived $C_{ant}$ concentrations that are not without uncertainties, particularly in the deep Arctic Ocean where measured CFC-12 concentrations are small. The high-resolution model would have sim-

ulated a higher Arctic $C_{ant}$ inventory had computational resources been available to use it throughout the entire industrial era rather than initializing it in 1958 with results from the intermediate-resolution model (ORCA05), in which the penetration of CFC-12 and $C_{ant}$ into Arctic intermediate waters is weaker. The largest source of differences in the $C_{ant}$ inventory between resolutions is due to the increasing ventilation of intermediate waters as model resolution is refined, as revealed by CFC-12 and $C_{ant}$ model–data comparison. The highest-resolution model, ORCA025, still underestimates the $C_{ant}$ data-based estimates at around 400 m and slightly overestimates them at around 1300 m. The deeper overestimate appears due to excessive penetration of $C_{ant}$-rich Atlantic water. The shallower underestimate may be partly due to inadequate representation of ventilation of intermediate waters via downslope flows that are driven by brine formation over the Arctic's enormous continental shelf, a transport process that is notoriously difficult to represent in *z*-coordinate models, especially at a lower resolution.

Our forced ocean simulations suggest that Arctic Ocean storage of $C_{ant}$ is driven mostly by net lateral inflow, the total input of which is about 3 times that of the air–sea flux. That 3 : 1 ratio varies little with resolution because both the lateral flux and the air–sea flux increase as resolution is refined. The lateral flux is typically less dominant in the CMIP5 models but its magnitude varies greatly as does its ratio relative to the air–sea flux. Some CMIP5 models even simulate net lateral outflow of $C_{ant}$, but those models also simulate unrealistically low $C_{ant}$ inventories. The only CMIP5 models that succeed in reaching the lower limit of the data-based $C_{ant}$ inventory range are those that have a large net lateral input. Unfortunately, the causes of the CMIP5 model differences remain unclear as is often the case when comparing models having many differences. Most of the CMIP5 models appear not to have been evaluated in terms of their ability to simulate realistic lateral water transport at the boundaries of the Arctic Ocean, which is fundamental to simulating realistic $C_{ant}$ but may be problematic given their coarse resolution. The next phase of CMIP is ongoing and includes CFC-12 and related transient tracers, which will help weigh simulated results for $C_{ant}$.

As the mass of simulated anthropogenic carbon in the Arctic Ocean increases with resolution, so does the simulated acidification. For instance, during 1960–2012, the average ASH in the Arctic shoals 4 times faster in ORCA025 than in ORCA2. Higher resolution is also needed to capture local extremes. Despite these benefits, the higher computational costs of making centennial-scale, high-resolution, biogeochemical ocean simulations remain prohibitive. More practical in the short term would be to assess effects from less costly model improvements, including heightened vertical resolution, subgrid-scale parameterizations, and adjustments to model parameters for viscosity and slip conditions. For such regional studies, nested models would offer the advantage of focused higher resolution while still avoiding adverse effects from imposed lateral boundary conditions. These efforts along with including more coastal ocean processes in global models should eventually lead to greater prognostic skill and more reliable projections not only for the Arctic Ocean but for regional seas and the coastal ocean in general.

*Code availability.* The code for the NEMO ocean model version 3.2 is available under CeCILL license at http://www.nemo-ocean. eu TS5.

### Appendix A: Perturbation vs. full biogeochemical approach

To assess the reliability of the perturbation approach, we compared its results from the coarse-resolution ocean model over 1870–2012 (P1870-ORCA2*), i.e., without branching from ORCA05, to those from the analogous full biogeochemical simulation (B1870-ORCA2*). Globally, the simulated $C_{ant}$ inventory with the perturbation approach in 2012 is 2 % larger than that with the full biogeochemical approach (Table A1). These differences are mainly located in the top 200 m (Fig. A1) of the tropics and the Southern Ocean (Fig. A2), where regional inventories of $C_{ant}$ from the perturbation approach overestimate those from the full approach by up to 3 %. Those two regions are also the ones that store most of the anthropogenic carbon. Conversely, in the Arctic, the perturbation approach underestimates the 2012 $C_{ant}$ inventory of the full approach by 3 % because of its deficit between 200 and 600 m, the depth zone that is directly affected by Atlantic inflow (Sect. 3.1.3). Overall, these differences are small, thus supporting our use of the more efficient perturbation approach to correct for the late start date of the full biogeochemical simulations.

These differences, although small, merit an explanation. The perturbation approach is regionally biased because its preindustrial reference state is assumed to be in equilibrium with the atmosphere everywhere (Sect. 2.3). Hence its results will differ from the full approach, which allows for disequilibrium between preindustrial atmospheric and oceanic $pCO_2$. For example, with the full approach, simulated surface-ocean $pCO_2$ in the tropics and Southern Ocean generally exceeds atmospheric $pCO_2$ under preindustrial conditions, a supersaturation that is also seen with ocean inversions for the same regions (Gruber et al., 2009). So by assuming equilibrium and not accounting for this supersaturation, the perturbation approach relies on a buffer capacity that is too high. That is, when its preindustrial surface-ocean $pCO_2$ reference is too low, its corresponding carbonate ion concentration is too high and thus so must be its buffer capacity, i.e., its chemical capacity to absorb $C_{ant}$.

**Table A1.** $C_{ant}$ inventories in 2012 in Pg C.

|  | Global Ocean | Arctic Ocean |
| --- | --- | --- |
| B1870-ORCA2* | 137.3 | 1.48 |
| P1870-ORCA2* | 139.6 | 1.43 |

In contrast, in the North Atlantic, surface-ocean $pCO_2$ is generally undersaturated in the full approach under preindustrial conditions (B1870-ORCA2* in 1870), as it is in ocean inversions (Gruber et al., 2009). By not accounting for this undersaturation, the perturbation approach overestimates the preindustrial surface-ocean $pCO_2$ and thus underestimates the corresponding reference carbonate ion concentration, buffer capacity, and uptake of $C_{ant}$ relative to the full approach. The growing influence of this underestimated uptake in the North Atlantic can be seen as its waters invade the Arctic during the course of the simulation (Fig. A3). That lateral invasion overwhelms the small but opposite tendency early in the perturbation simulation to overestimate Arctic $C_{ant}$ uptake, an artifact of the perturbation approach's preindustrial reference state not accounting for local impacts from riverine inputs. Conversely, in the full approach with PISCES, riverine inputs typically lower the carbonate ion concentration and buffer capacity of shelf seas.

Despite its simplifications, the perturbation approach differs little from the full approach in terms of basin-wide results. With it, we can envision garnering sufficient computational resources to soon make a global $C_{ant}$ simulation at high resolution (ORCA025) over the full industrial era without branching it off from a lower-resolution model along the way. That should in turn allow us to help further refine limits for $C_{ant}$ uptake and storage in the Arctic as well as other regions. With the full biogeochemical approach, this would not be feasible for years to come.

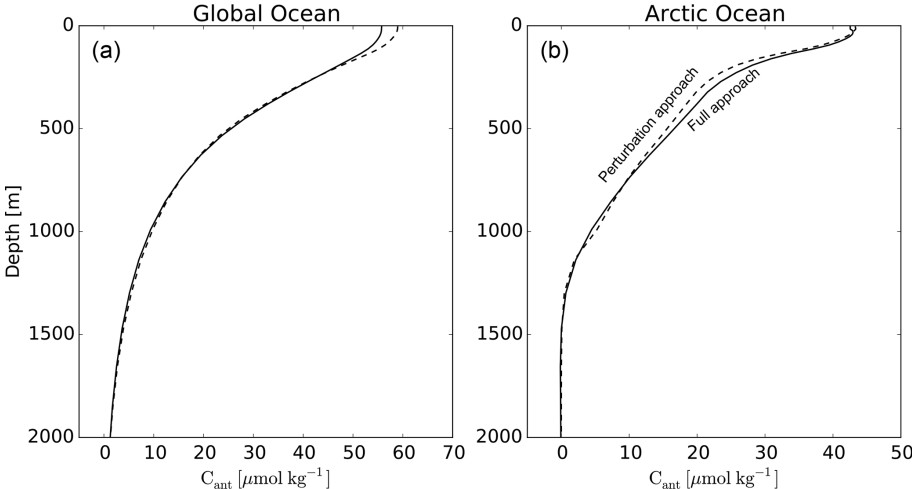

**Figure A1.** Mean vertical profiles of $C_{ant}$ for the global ocean **(a)** and the Arctic Ocean **(b)** in 2012 for the full biogeochemical approach (B1870-ORCA2*) (solid) and the perturbation approach (P1870-ORCA2*) (dashed).

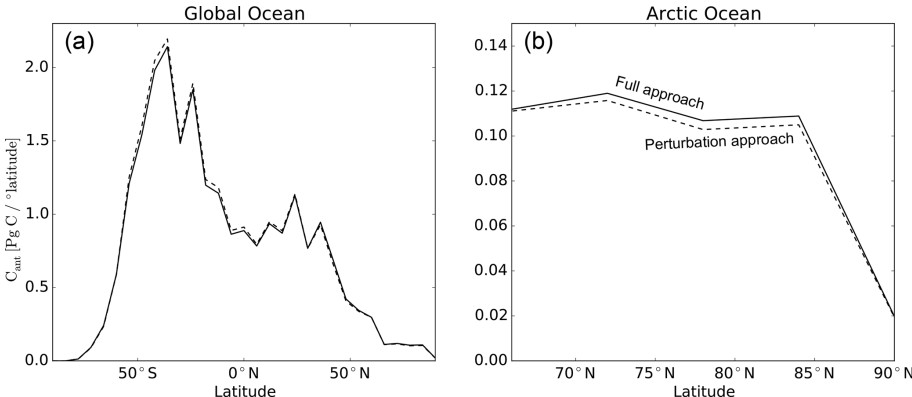

**Figure A2.** Zonal integral of vertically integrated $C_{ant}$ per degree of latitude in 2012 for the global ocean **(a)** and the Arctic Ocean **(b)** using the full biogeochemical approach (B1870-ORCA2*) (solid) and the perturbation approach (P1870-ORCA2*) (dashed)

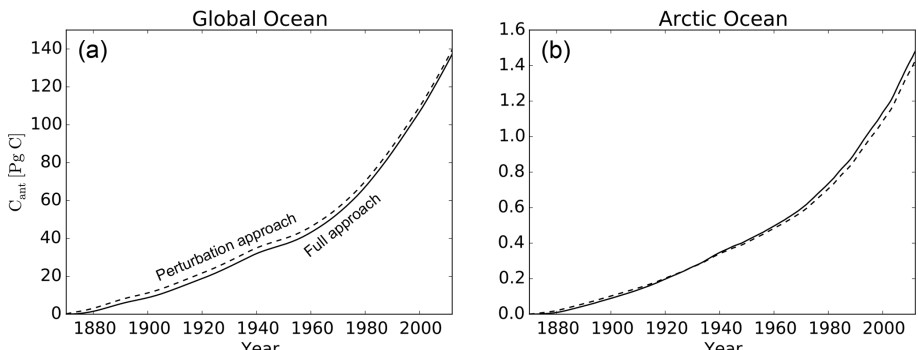

**Figure A3.** Arctic Ocean $C_{ant}$ inventory in the global ocean **(a)** and the Arctic Ocean **(b)** for the full biogeochemical approach (B1870-ORCA2*) (solid) and the perturbation approach (P1870-ORCA2*) (dashed).

*Supplement.* The supplement related to this article is available online at: https://doi.org/10.5194/bg-16-1-2019-supplement.

*Author contributions.* .TS6

*Competing interests.* The authors declare that they have no conflict of interest.

*Acknowledgements.* We thank TS7 J.-M. Molines, L. Brodeau, and B. Barnier for developing the DRAKKAR ORCA05 and ORCA025 global configurations of NEMO and J. Simeon for the implementations of those configurations and ORCA2L46 with PISCES. The research was funded by the EU H2020 project C-CASCADES (Marie Sklodowska-Curie grant 643052). The coauthors also acknowledge support from the EU H2020 CRESCENDO project (grant 641816), the ANR SOBUMS project (ANR-16-CE01-0014), and the MTES Acidoscope project. Simulations were made using HPC resources from the French GENCI-IDRIS program (grant x2015010040). Model output was stored and analyzed on the Ciclad platform at IPSL. CE22

*Review statement.* This paper was edited by Marilaure Grégoire and reviewed by two anonymous referees.

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

## Remarks from the language copy-editor

## Remarks from the typesetter