# Peer review of "Model constraints on the anthropogenic carbon budget of the Arctic Ocean"

_Biogeosciences, 2018_

## Referee Comment (RC1) · Anonymous Referee #1 · 1 Aug 2018

**1   Overview**

Terhaar et al. ask: what effect does model resolution have on simulated Arctic Ocean anthropogenic $CO_2$ storage and acidification? The answer: increased model resolution shows higher regional storage by up to 25%, moving the inventory closer to data-based estimates, and increased acidification with faster shoaling of the aragonite saturation horizon.

This is an interesting and useful question, and the study has been well-designed to answer it. The results and their interpretation seem sensible, although as mentioned below, a robust uncertainty analysis is critically lacking.

The manuscript is interesting and easy to read. The Introduction is very well-written and Methods are clear. Results and Discussion are succinct, but in places the Discussion in particular could be developed further to provide more insight. Many of the questions below are really prompts in this direction.

**2   Scientific questions**

The highest-resolution model is "still not eddy resolving" (3-28 i.e. page 3, line 28). Does that mean that you would expect further changes still with yet higher resolution? Would you expect the anthropogenic $CO_2$ inventory to increase even more?

The monthly averaging process introduces an error of 27% for the lowest-resolution model (7-20). This is a similar magnitude to the difference in anthropogenic $CO_2$ inventory between the different resolutions, which is your most important result. Does this error being the same size as the 'signal' not significantly reduce your confidence in the results (i.e. differences between resolutions)?

Following from the previous point, what is the uncertainty in the differences between your inventories at different resolutions? Without a robust estimate of this we cannot trust that they are meaningfully different. This would also aid comparisons with data-based estimates (e.g. in Section 4.3). Section 2.6: the recent decline in atmospheric CFC-12 is problematic for applications of the TTD approach after the concentration peak. Other still-increasing tracers such as $SF_6$ may be more suitable for this recent period. Could this adversely influence your results, and if so are alternative tracer data available for comparison? Might this be linked to why your simulated CFC-12 underestimates the observations in the upper water column (Section 4.1)?

You note that overestimation of sea-ice cover should reduce air-sea $CO_2$ exchange (9-14). There are a number of observational studies that attempt to quantify this effect. Can these be used to quantify your statement?

Section 3.4: my impression was that the primary reason to expect model resolution to influence the anthropogenic $CO_2$ inventory was because of better representation of circulation features. Therefore the increase in lateral flux, being a function of circulation, is expected – but the simultaneous increase in the air-sea flux does not seem so intuitive. Indeed if additional anthropogenic $CO_2$ is being transported into the region from elsewhere we might expect this to increase total dissolved inorganic carbon and thus reduce net air-to-sea $CO_2$ flux. Do you have a conceptual explanation for what is driving the air-sea flux increase with resolution?

In order to declare that two things are 'not statistically different' (14-12) you must also provide the statistical information that were used to demonstrate this.

Does the increase in resolution alter lateral anthropogenic $CO_2$ fluxes primarily because of the representation of circulation (1) inside the Arctic Ocean, (2) at its boundaries/interfaces, (3) in the non-Arctic global ocean, or (4) everywhere? You note that for computational reasons we cannot routinely run these models globally at high resolution, but if only one region of the model needed to be at high resolution to achieve your results, would it be possible to strike a balance with a hybrid resolution model?

Some of the notes about possible future work on CFC-12 and the TTD parameters in the Conclusions would probably be more suited to the Discussion.

No other studies have been mentioned that have attempted to answer this same question for the Arctic Ocean, but there have been other investigations of the effect of increased model resolution in various contexts. Do these provide any insights that would be useful in interpreting your results?

**3  Figures**

Use of red vs green (e.g. Fig. 3) with no difference in line style can render these lines indistinguishable to colorblind readers.

The blue-green-yellow-red color scale used on transect plots (e.g. Fig. 4) is not perceptually uniform, leading to visual artifacts such as false boundaries.

Depth should be positive going down into the ocean (Figs. 4 and 6).

**4  Technical/grammatical notes**

There is inconsistency in usage of past and present tenses in the Methods.

The contexts in which the word 'though' is used are highly colloquial and, to me, not suited for scientific writing (2-12, 14-29, 15-8).

Suggested corrections:

| | |
|---|---|
| 1-6 | eddy-admitting |
| 2-3 | consequences for |
| 4-30 | following Moore et al. |
| 10-24 | (Fig. 6) ? |
| 11-21 | Arctic Ocean basins |
| 15-33 | reword this sentence to indicate the direction of the effect |

---

## Referee Comment (RC2) · Anonymous Referee #2 · 18 Sep 2018

**Comments on "Model constraints on the anthropogenic carbon budget of the Arctic Ocean"**

by Terhaar et al., submitted to Biogeosciences, doi: 10.5194/bg-2018-283

In this study the authors examine the anthropogenic $CO_2$ budget of the Arctic Ocean and how this inventory depends on model resolution. In that purpose they take advantage of the NEMO-v3.2 OGCM coupled to the biogeochemical model PISCES-v1. They perform experiments with three different horizontal resolution of the OGCM, namely 2°, 0.5°, and 0.25°. Inventories of anthropogenic carbon in the Arctic appear to increase with increasing resolution (from 2.0 to 2.6 Pg C). The role of air-sea fluxes and lateral transport in building these inventories is examined. In this model lateral transport accounts for 75% of the Arctic Ocean anthropogenic $CO_2$ inventory. A comparative study of the outputs of other modeling studies (CMIP5) allow concluding that models with larger lateral transport appear to better fit data-based estimates of the anthropogenic carbon in the Arctic Ocean. This partitioning does not depend on the model resolution. Resolution appears important in shaping the tracer distribution and improving data-model agreement.

The paper is well written and very well structured. However I have several concerns about the method and the way data-model comparison is performed. Before the method is thoroughly assessed this paper is not fit for publication.

**Major comments**

1.  I have serious concerns about the applied method for estimating the $C_{ant}$. Conclusions about the impact of model resolution might not be robust due to shortcoming in the method.

    $C_{ant}$ is rightly defined as the difference between the simulated historical and control ocean dissolved carbon contents. However, there is only one control experiment performed (page 5), that for ORCA05. As far as I understand the $C_{ant}$ for ORCA2 and ORCA025 is evaluated as the difference between the respective 1958→2012 experiments and the ORCA05 control for the same period. Therefore I strongly suspect that the differences in CFC and $C_{ant}$ among the different models may be explained by model drift.

    In order to lift that concern the following actions should be taken:

    a.  Perform control experiments over the period 1958→2012 for each model resolution.

    b.  While it is defensible to reduce the computation length with the high resolution model (ORCA025) there is no such need for ORCA2, which runs even faster than ORCA05. The authors should also present results of historical and control experiments performed with ORCA2. The perturbation experiments should also be repeated with ORCA2.

    The results of these additional experiments should then be compared to those presented in the present paper. This would provide a means of validating their method and assessing potential drifts

2.  The experiments which are presented here are global. What would be the global figures for anthropogenic CO2 uptake in the 5 cases? How do these figures compare to other

assessments? Answering this request would allow evaluating whether the OGCMs as a whole would need serious refinements or should the effort concentrate on less-well resolved areas such as the Arctic Ocean.

3. The other main concern deals with the correction of data-based reconstructions of $C_{ant}$ (Abstract, Sections 4.2 and 4.5, Fig. 9). The authors assume that reconstructed deep values of $C_{ant}$ should be corrected downwards since observed CFC-12 concentrations at those depths are negligible. Doing so means overlooking the important fact that CFCs started to be emitted in the atmosphere much later than $CO_2$. Data-based estimates relying on the TTD method take into account the different tracer histories in the atmosphere. Clearly, the TTD method has limitations. The end-product displays rather large uncertainties. However, there are no sound arguments for setting the $C_{ant}$ in the deep Arctic to zero.

4. Modeled CFC-12 inventories in the Arctic (Fig. 5 and page 12, lines 24 and 25) appear to be much lower than the observed ones, even with ORCA25. Would it be possible to provide total (integrated over depth and distance) inventories along the AOS94 and Beringia 2005 expedition pathways and compare the 3 model results to the data inventories? The low CFC inventory provides an indication that low $C_{ant}$ would be expected too.

5. In addition, a description of how CFC-12 is modeled is lacking.

**Minor comments**

- Abstract, line 10: $C_{ant}$ is not defined yet.

- Page 3, line 2: a reference to the figure displaying the map of the Arctic should be made here; the reader does not necessarily know about the area characteristics. In this sense Fig. 2 should become Fig. 1.

- Page 3, line 2: "The bathymetry of the Arctic Ocean differs from that of the   other oceans..."

- Page 3, line 25: is 'laminar' right?

- Page 4, line 2: table 3 does not come into order.

- Page 4, lines 11 and 12: "NEMO uses partial steps so that the model better matches the observed topography. Thus the depth of the deepest cell can be smaller than the original grid cell." Could you develop or reformulate? It is hard to understand what it is meant here.

- Page 4, line 22: there is no mention of the Si:P and Fe:P ratios.

- Page 4, line 29: does sediment mobilization only intervene in the Fe cycle? Or does it also affect the other nutrients?

- Page 4, line 30: "… following  Moore et al. (2004)."

- Page 5, line 33: "… simulations made  with the same circulation model..."

- Pages 5 and 6: the many occurrences of 'xCO$_2$' should be changed into 'CO$_2$'.

- Page 6, equation (1): what are the units of pCO2 and T?

- Page 6, line 25: "given that it is based on results from ORCA05."

- Page 6, line 28: reference to Table 4 should appear here.

- Page 10, line 24: "apparent in ORCA025 ."

- Page 12, line 22: "that  excess simulated CFC-12 between 1000 and 2000 m..."

- Tables do not come into order. Table 3 should become Table 1, Table 1 → 2, and Table 2 → 3.

- Table 1: the 'b' subscript does not appear anywhere in the table

- Table 2, caption: "Fitted parameters for the perturbation approach s starting in 1765 (P1765) and in 1870 (P1870)."

- Table 4: what do exactly represent the lines "Total transport" and "Summed lateral flux"?

- Table 4, caption: "Simulated values are calculated for the same time period as observations."

- Fig. 1 and Fig. 2 should be inverted

- Fig. 10, caption:

  - The first sentence "Profiles of $\Omega_A$ ." is confusing. I suggest to remove most of it; it is not needed.

  - "Results are shown for ORCA05 in 1960 (black solid) as well as **for** ORCA2 (green dot-dash), ORCA05 (red dashes), and ORCA025 (blue dots) in 2012.".

---

## Author Comment (AC1) · 9 Oct 2018

**Response to Referee #1**

We thank both referees for their comments and suggestions. The manuscript will be revised to thoroughly address each point. Generally the plan is as follows:

(1) The description of the simulations analyzed in our study will be improved by adding more details concerning (a) how the different simulations (ORCA2, ORCA05, and ORCA025) were made, making it clear that each of these has its own control simulation, (b) how CFC-12 was simulated, and (c) how carbon transport was estimated at

the boundaries of the Arctic Ocean.

(2) The discussion section will be expanded to provide more detail about (a) the effect of increasing resolution in ocean models found by other modeling studies and (b) the mechanisms influencing changes in air-sea $CO_2$ fluxes in the Arctic between the different resolutions, including a discussion of the role of sea-ice. Additional analysis will also be included to show how the different configurations compare to each other in terms of the global-ocean inventory of anthropogenic carbon.

During the review period, we also discovered an issue with our CFC-12 simulations (initialization to non-zero concentrations). Hence we have rerun all CFC-12 simulations (as will be detailed in the revised manuscript). Furthermore, we have used the opportunity to complement the ORCA05 $C_{ant}$ perturbation simulations with analogous simulations for the ORCA2 and ORCA025 configurations (each initialized in the beginning of 1958 with output from the last time step in 1957 of the ORCA05 $C_{ant}$ perturbation simulation and run until 2012). With these updated simulations, the model-data CFC-12 comparison has been improved (to be discussed in Sections 3.2 and 4.1) as have been the corrections for the estimated $C_{ant}$ fluxes at the boundaries (both lateral and at the air-sea interface). The figures and tables of the revised manuscript will be updated accordingly. Despite these improvements, the Conclusions of our study remain the same.

In the following we address their concerns point by point.

**Referee # 1**

1) Overview

Terhaar et al. ask: what effect does model resolution have on simulated Arctic Ocean anthropogenic $CO_2$ storage and acidification? The answer: increased model resolution shows higher regional storage by up to 25%, moving the inventory closer to data-based estimates, and increased acidification with faster shoaling of the aragonite saturation horizon. This is an interesting and useful question, and the study has been well-designed to answer it. The results and their interpretation seem sensible, although as mentioned below, a robust uncertainty analysis is critically lacking. The manuscript is interesting and easy to read. The Introduction is very well-written and Methods are clear. Results and Discussion are succinct, but in places the Discussion in particular could be developed further to provide more insight. Many of the questions below are really prompts in this direction.

**Reply**: Thank you very much. In the revised manuscript, we will strive to address these concerns in detail.

2) Scientific questions

**Reviewer Question 1** — The highest-resolution model is "still not eddy resolving" (3-28 i.e. page 3, line 28). Does that mean that you would expect further changes still with yet higher resolution? Would you expect the anthropogenic $CO_2$ inventory to increase even more?

**Reply**: Without actually making higher resolution simulations, one can of course never be certain. However, we would agree with Referee #1 that the anthropogenic carbon

inventory in the Arctic Ocean would probably increase if the model resolution were refined even further. The $C_{ant}$ inventory increases by 0.38 Pg C between ORCA2 and ORCA05 and by 0.27 Pg C between ORCA05 and ORCA025, so we may not have reached the nearly flat part of the asymptote. Further enhancement of resolution could further increase the inventory via the combined effects of improved bathymetry, increased coastal water velocities, and enhanced surface-to-deep transport of passive tracers from brine formation on Arctic shelves. Concerning the latter, even higher model resolution might well lead to higher $C_{ant}$ concentrations in the Canada basin and to refining the chimneys of higher CFC-12 concentrations in the Canada basin, which are observed but only barely resolved in ORCA025 (page 10, lines 23-24). We will add these considerations in section 4.4 of the revised manuscript.

**Reviewer Question 2** — The monthly averaging process introduces an error of 27% for the lowest-resolution model (7-20). This is a similar magnitude to the difference in anthropogenic $CO_2$ inventory between the different resolutions, which is your most important result. Does this error being the same size as the 'signal' not significantly reduce your confidence in the results (i.e. differences between resolutions)?

**Reply**: Our error estimate of 27% applies only to the ORCA2 transport of $C_{ant}$, calculated separately at each of the 4 boundaries of the Arctic Ocean from monthly-average model output (offline), an important point raised by Referee #1 for which we should have been clearer. That offline calculation error is smaller at higher resolution (e.g, 4% with ORCA025). Moreover, it does not apply to our estimates of $C_{ant}$ inventory, the cumulative air-sea $C_{ant}$ flux, and the total lateral flux, all of which are calculated online in the model and for which associated calculation errors are negligible.

This key point will be clarified in the revised manuscript.

**Reviewer Question 3** — You note that overestimation of sea-ice cover should reduce

air-sea $CO_2$ exchange (9-14). There are a number of observational studies that attempt to quantify this effect. Can these be used to quantify your statement?

**Reply**:  Our overestimation of sea-ice extent is less than 3%. Thus it is of second order when estimating the $C_{ant}$ air-sea flux. There are studies that have estimated the effect of declining sea-ice on air-sea fluxes of total carbon (Bates et al. [2006], Cai et al. [2010]), but they do not agree on whether the flux will increase due to increasing biological activity or decrease due to higher stratification and increases in riverine organic carbon. Because these studies focus on the air-sea flux of total $CO_2$ (natural + anthropogenic), we cannot use them to extract the response of only the anthropogenic component, the focus of our study.

**Reviewer Question 4**  —  Section 3.4: my impression was that the primary reason to expect model resolution to influence the anthropogenic $CO_2$ inventory was because of better representation of circulation features. Therefore the increase in lateral flux, being a function of circulation, is expected – but the simultaneous increase in the air-sea flux does not seem so intuitive. Indeed if additional anthropogenic $CO_2$ is being transported into the region from elsewhere we might expect this to increase total dissolved inorganic carbon and thus reduce net air-to-sea $CO_2$ flux. Do you have a conceptual explanation for what is driving the air-sea flux increase with resolution?

**Reply**:  Two mechanisms may explain the increase in the air-sea flux with resolution: (1) higher resolution increases the amount of $C_{ant}$ that is advected into the Arctic Ocean through the Fram Strait via subsurface currents, which does not substantially affect surface $C_{ant}$ concentrations nor hence air-sea exchange of anthropogenic carbon and (2) higher resolution enhances deep-water formation in the Arctic Ocean , mainly in the Barents Sea as shown in the CFC-12 profiles (Figure 7), which reduces surface $C_{ant}$ and thus enhances the air-to-sea flux of anthropogenic carbon.

We will add this explanation to Section 4.4 of the revised manuscript.

**Reviewer Question 5** — In order to declare that two things are 'not statistically different' (14-12) you must also provide the statistical information that were used to demonstrate this.

**Reply**: In the revised manuscript, the corresponding sentence will be changed to 'The simulated air-sea flux falls within that assigned uncertainty range for the data-based estimate.'

**Reviewer Question 6** — Does the increase in resolution alter lateral anthropogenic $CO_2$ fluxes primarily because of the representation of circulation (1) inside the Arctic Ocean, (2) at its boundaries/interfaces, (3) in the non-Arctic global ocean, or (4) everywhere?

**Reply**: We tried to address this question in Section 4.4 of the submitted manuscript, but we only discuss circulation differences inside the Arctic Ocean and at its boundaries. We showed that increasing resolution from ORCA2 to ORCA05 affects mainly the circulation at the boundaries (1) while the change from ORCA05 to ORCA025 affects mainly circulation inside the Arctic Ocean (2).

For more insight into the role of the global ocean (outside the Arctic), we have now calculated the total $C_{ant}$ inventory for the global ocean with all three resolutions (see below). Globally the three resolutions agree within 3%. This agreement is much tighter than that for the Arctic Ocean $C_{ant}$ inventory. This new comparison suggests a weak role of the non-Arctic global ocean on the Arctic Ocean $C_{ant}$ budget (3)(4), although resolution-dependent changes in regions adjacent to the Arctic Ocean's lateral boundaries (e.g., in the North Atlantic) may well have an effect. These considerations will be added to section 4.4 of the revised manuscript.

**Reviewer Question 7** — You note that for computational reasons we cannot routinely run these models globally at high resolution, but if only one region of the model needed

to be at high resolution to achieve your results, would it be possible to strike a balance with a hybrid resolution model?

**Reply**: Certainly such approaches, using for example the nesting tool of NEMO (AGRIF, Debreu et al. [2008]) as in Duteil et al. [2014] would be an interesting option. Unfortunately, this nesting tool is known to not work well in ice-covered regions nor with the biogeochemical model PISCES. Hence we leave this for future work as will be mentioned in the Conclusion of the revised manuscript.

**Reviewer Question 8** — Some of the notes about possible future work on CFC-12 and the TTD parameters in the Conclusions would probably be more suited to the Discussion.

**Reply**: We are considering to move these notes to the Discussion.

**Reviewer Question 9** — No other studies have been mentioned that have attempted to answer this same question for the Arctic Ocean, but there have been other investigations of the effect of increased model resolution in various contexts. Do these provide any insights that would be useful in interpreting your results?

**Reply**: In terms of ocean biogeochemistry, most studies on the effect of resolution have focused on the impacts of mesoscale to submesoscale structures on phytoplankton and productivity (e.g., Lévy [2008], McGillicuddy [2016]). Other studies have investigated the role of increased model resolution on transient tracers (Lachkar et al., [2007]) and on carbon and oxygen (Duteil et al., [2014]), but they have focused on other regions (Southern Ocean and the tropics). Therefore it is difficult to transfer their findings to the Arctic Ocean. Some discussion on other studies focusing on the role of model resolution will be include in the Discussion of the revised manuscript.

3) Figures

**Reviewer Question 10** — Use of red vs green (e.g. Fig. 3) with no difference in line style can render these lines indistinguishable to colorblind readers.

**Reviewer Question 11** — The blue-green-yellow-red color scale used on transect plots (e.g. Fig. 4) is not perceptually uniform, leading to visual artifacts such as false boundaries.

**Reviewer Question 12** — Depth should be positive going down into the ocean (Figs. 4 and 6).

**Reply**: These suggestions for improving the style and colors will be implemented in the revised manuscript.

4) Technical/grammatical notes

**Reviewer Question 13** — There is inconsistency in usage of past and present tenses in the Methods.

**Reply**: The Methods section will be improved to avoid this inconsistency in the revised manuscript.

**Reviewer Question 14** — The contexts in which the word 'though' is used are highly colloquial and, to me, not suited for scientific writing (2-12, 14-29, 15-8).

**Reply**: We will avoid the use of the word 'though' in the revised manuscript.

**Reviewer Question 15** — Suggested corrections:
1-6   eddy-admitting
2-3   consequences for
4-30   following Moore et al.
10-24 (Fig. 6) ?
11-21 Arctic Ocean basins
15-33 reword this sentence to indicate the direction of the effect

**Reply**: The manuscript will be revised according to these suggestions. For 15-33 we will write "Thus model resolution also affects the time at which waters become under-saturated with respect to aragonite with higher resolutoin producing greater shoaling."

References

Bates, N. and Mathis, J. (2009). The arctic ocean marine carbon cycle: evaluation of air-sea co 2 exchanges, ocean acidification impacts and potential feedbacks. Biogeosciences, 6(11):2433– 2459. 8

Cai, W.-J., Chen, L., Chen, B., Gao, Z., Lee, S. H., Chen, J., Pierrot, D., Sullivan, K., Wang, Y., Hu, X., et al. (2010). Decrease in the co2 uptake capacity in an ice-free arctic ocean basin. Science, 329(5991):556–559. 2

Debreu, L., Vouland, C. and Blayo, E. AGRIF: Adaptive grid refinement in Fortran. Computers Geosci. 34, 8–13 (2008).

Duteil, O., F. U. Schwarzkopf, C. W. Böning, and A. Oschlies (2014), Major role of the equatorial current system in setting oxygen levels in the eastern tropical Atlantic Ocean: A high‐resolution model study, Geophys. Res. Lett., 41, 2033–2040, doi: 10.1002/2013GL058888.

Lachkar, Z., Orr, J. C., Dutay, J.-C., and Delecluse, P.: Effects of mesoscale eddies on global ocean distributions of CFC-11, CO2, and $\Delta$14C, Ocean Sci., 3, 461-482, https://doi.org/10.5194/os-3-461-2007, 2007.

Lévy M. (2008) The Modulation of Biological Production by Oceanic Mesoscale Turbulence. In: Weiss J.B., Provenzale A. (eds) Transport and Mixing in Geophysical Flows. Lecture Notes in Physics, vol 744. Springer, Berlin, Heidelberg

McGillicuddy Jr, D. J. (2016). Mechanisms of physical-biological-biogeochemical interaction at the oceanic mesoscale.

---

## Author Comment (AC2) · 9 Oct 2018

**Response to Referee #2**

We thank both referees for their comments and suggestions. The manuscript will be revised to thoroughly address each point. Generally the plan is as follows:

(1) The description of the simulations analyzed in our study will be improved by adding more details concerning (a) how the different simulations (ORCA2, ORCA05, and ORCA025) were made, making it clear that each of these has its own control simulation, (b) how CFC-12 was simulated, and (c) how carbon transport was estimated at

[Figure]

the boundaries of the Arctic Ocean.

(2) The discussion section will be expanded to provide more detail about (a) the effect of increasing resolution in ocean models found by other modeling studies and (b) the mechanisms influencing changes in air-sea $CO_2$ fluxes in the Arctic between the different resolutions, including a discussion of the role of sea-ice. Additional analysis will also be included to show how the different configurations compare to each other in terms of the global-ocean inventory of anthropogenic carbon.

During the review period, we also discovered an issue with our CFC-12 simulations (initialization to non-zero concentrations). Hence we have rerun all CFC-12 simulations (as will be detailed in the revised manuscript). Furthermore, we have used the opportunity to complement the ORCA05 $C_{ant}$ perturbation simulations with analogous simulations for the ORCA2 and ORCA025 configurations (each initialized in the beginning of 1958 with output from the last time step in 1957 of the ORCA05 $C_{ant}$ perturbation simulation and run until 2012). With these updated simulations, the model-data CFC-12 comparison has been improved (to be discussed in Sections 3.2 and 4.1) as have been the corrections for the estimated $C_{ant}$ fluxes at the boundaries (both lateral and at the air-sea interface). The figures and tables of the revised manuscript will be updated accordingly. Despite these improvements, the Conclusions of our study remain the same.

In the following we address their concerns point by point.
**Referee # 2**

In this study the authors examine the anthropogenic $CO_2$ budget of the Arctic Ocean and how this inventory depends on model resolution. In that purpose they take advantage of the NEMO-v3.2 OGCM coupled to the biogeochemical model PISCES-v1. They perform experiments with three different horizontal resolution of the OGCM, namely 2°, 0.5°, and 0.25°. Inventories of anthropogenic carbon in the Arctic appear to increase with increasing resolution (from 2.0 to 2.6 Pg C). The role of air-sea fluxes and lateral transport in building these inventories is examined. In this model lateral transport accounts for 75% of the Arctic Ocean anthropogenic $CO_2$ inventory. A comparative study of the outputs of other modeling studies (CMIP5) allow concluding that models with larger lateral transport appear to better fit data-based estimates of the anthropogenic carbon in the Arctic Ocean. This partitioning does not depend on the model resolution. Resolution appears important in shaping the tracer distribution and improving data-model agreement.

The paper is well written and very well structured. However I have several concerns about the method and the way data-model comparison is performed. Before the method is thoroughly assessed this paper is not fit for publication.

Major comments

**Reviewer Question 1** — I have serious concerns about the applied method for estimating the $C_{ant}$. Conclusions about the impact of model resolution might not be robust due to shortcoming in the method.

$C_{ant}$ is rightly defined as the difference between the simulated historical and control ocean dissolved carbon contents. However, there is only one control experiment performed (page 5), that for ORCA05. As far as I understand the $C_{ant}$ for ORCA2

and ORCA025 is evaluated as the difference between the respective 1958→2012 experiments and the ORCA05 control for the same period. Therefore I strongly suspect that the differences in CFC and $C_{ant}$ among the different models may be explained by model drift. In order to lift that concern the following actions should be taken:

a. Perform control experiments over the period 1958→2012 for each model resolution.

b .While it is defensible to reduce the computation length with the high-resolution model (ORCA025) there is no such need for ORCA2, which runs even faster than ORCA05. The authors should also present results of historical and control experiments performed with ORCA2. The perturbation experiments should also be repeated with ORCA2.

The results of these additional experiments should then be compared to those presented in the present paper. This would provide a means of validating their method and assessing potential drifts

**Reply**: a. We realize now that our original manuscript is woefully unclear about this point. We did not properly convey what was done. Indeed, each of our three resolutions already has its own control experiment over 1958-2012. For each resolution, $C_{ant}$ was computed from two simulations (historical and control), both made at the same resolution. Therefore, there is no resolution-related drift issue. This point will be clarified in section 2.2 of the revised manuscript.

b. As pointed out, we had already made a control simulation in ORCA2-PISCES from 1958 to 2012 although that was not clear in the submitted manuscript. For the same period, we have in addition added a $C_{ant}$ perturbation simulation in ORCA2. Our strategy to consistently use the same ORCA05 output from the end of 1957 to initialize

[Figure]

ORCA2 as well as ORCA025 in 1958 was by design. It produces a consistent set of results whereby the effect of resolution can be compared rigorously. Had we started all ORCA-PISCES simulations in 1870, which was not computationally feasible, there would have been larger differences due to resolution, e.g., based on the divergence shown in Figure 1. Hence the differences due to resolution discussed in the submitted manuscript are probably a lower limit, something that will be clarified in the revised manuscript. In addition, for the revised manuscript, we are considering making an additional $C_{ant}$ perturbation simulation in ORCA2 initialized in 1765 (and 1870), if allowed by our limited computational resources available at the end of this fiscal year. That new simulation would provide a more complete comparison with the analogous reference ORCA05 simulation as suggested by Referee #2.

**Reviewer Question 2** — The experiments, which are presented here are global. What would be the global figures for anthropogenic $CO_2$ uptake in the 5 cases? How do these figures compare to other assessments? Answering this request would allow evaluating whether the OGCMs as a whole would need serious refinements or should the effort concentrate on less-well resolved areas such as the Arctic Ocean.

**Reply**: The global corrected $C_{ant}$ inventories for 1765-2009 for the three resolutions are given below as are the uncorrected inventories for 1870-2009 (in parentheses):

- ORCA2: 155 (130) Pg C

- ORCA05: 149 (124) Pg C

- ORCA025: 151 (127) Pg C

These corrections were made by adding the difference between the two tracers in the $C_{ant}$ perturbation simulations, one initialized to zero in 1765 and the other in 1870, in each of the three resolutions. Those global perturbation results are as follows:

- ORCA2: 130 Pg C/155 Pg C

- ORCA05: 127 Pg C/152 Pg C

- ORCA025: 120 Pg C/144 Pg C

In the revised manuscript then, we will present the $C_{ant}$ inventory for 9 cases rather than 5 because of the new perturbation simulations in ORCA2 and ORCA025 (4 additional cases since each has 2 perturbation tracers: one initialized in 1765 and the other in 1870) that were not provided in the previously submitted version of the manuscript.

Regarding other assessments, Khatiwala et al. (2009) report a data-based estimate for the global $C_{ant}$ inventory for the period from 1765-2009 to be $140 \pm 24$ Pg C. For that same period, our results all lie within that range, falling near the upper boundary. Given the agreement of model results and data-based estimates for the global ocean $C_{ant}$ budget, it does not appear that further enhancements to resolution are needed to improve the global carbon budget. However, we would expect that improving resolution will have a large impact on some regional budgets, e.g., in zones where the ratio between the areas shelf seas vs. open ocean is relatively large, such as in the Arctic Ocean. Inventories of anthropogenic carbon in regions with small areal extent will have little impact on the global inventory, but they do provide some indicator of the potential enhanced effect of ocean acidification in those regions. These results and concerns will be brought up in section 4.2 of the revised manuscript.

**Reviewer Question 3** — The other main concern deals with the correction of data-based reconstructions of $C_{ant}$ (Abstract, Sections 4.2 and 4.5, Fig. 9). The authors assume that reconstructed deep values of $C_{ant}$ should be corrected downwards since observed CFC-12 concentrations at those depths are negligible. Doing so means over-looking the important fact that CFCs started to be emitted in the atmosphere much later

than $CO_2$ .. Data-based estimates relying on the TTD method take into account the different tracer histories in the atmosphere. Clearly, the TTD method has limitations. The end-product displays rather large uncertainties. However, there are no sound arguments for setting the $C_{ant}$ in the deep Arctic to zero.

**Reply**: Actually, there is some evidence that the GLODAPv2 estimate using the TTD method may overestimate $C_{ant}$ concentrations in the deep Arctic Ocean. First, the water mass mean ages of deep and bottom waters are estimated to be about 250 to 300 years in the Eurasian basins (Nansen and Amundsen basins) and around 450 years in the Canadian basin (Makarov and Canada basins) (Tanhua et al., [2009]; Schlosser et al., [1994]). Thus one would expect very little if any $C_{ant}$ would have reached those old deep waters. Second, the TTD method is known to estimate $C_{ant}$ concentrations around 5 $\mu$molkg$^{-1}$ even when the CFC-12 concentrations approach zero (Waugh et al. [2006]), which demonstrates the lack of sensitivity and large uncertainty associated with the TTD estimates for older water masses. Given that data-based $C_{ant}$ concentrations in the old, deep Arctic Ocean water masses are 18% of surface concentrations while CFC-12 concentrations in the same deep water masses are only 3% of surface concentrations, it is plausible that the TTD method overestimates $C_{ant}$.

In the revised manuscript, we will rephrase the text so as to indicate that to calculate the maximum potential error in the TTD-based estimate for the $C_{ant}$ inventory in the Arctic, we also set the deep TTD estimates to zero.

**Reviewer Question 4** — Modeled CFC-12 inventories in the Arctic (Fig. 5 and page 12, lines 24 and 25) appear to be much lower than the observed ones, even with ORCA25. Would it be possible to provide total (integrated over depth and distance) inventories along the AOS94 and Beringia 2005 expedition pathways and compare the 3 model results to the data inventories? The low CFC inventory provides an indication that low $C_{ant}$ would be expected too.

**Reply**:  Excellent suggestion. Thank you. We have now calculated these total inventories integrated over depth and distance along sections:

- Beringia 2005

  - Observations: 9.4 $\mu$molm$^{-1}$
  - ORCA025: 7.7 $\mu$molm$^{-1}$
  - ORCA05: 5.8 $\mu$molm$^{-1}$
  - ORCA2: 3.7 $\mu$molm$^{-1}$

- AOS94

  - Observations: 5.5 $\mu$molm$^{-1}$
  - ORCA025 : 4.8 $\mu$molm$^{-1}$
  - ORCA05 : 3.5 $\mu$molm$^{-1}$
  - ORCA2 : 2.0 $\mu$molm$^{-1}$

For both expeditions, the observed CFC-12 section inventories are underestimated by 13-18% in ORCA05, 31-38% in ORCA05, and 61-64% in ORCA2. This tendency with resolution for these section inventories is consistent with that seen for the Arctic Ocean's basin-wide inventory where the data-based estimate is underestimated by 13-15% in ORCA025, 22-24% in ORCA05, and 34-36% in ORCA2.

A table of these integrated values and corresponding text will be included in Section 3.2 of the revised manuscript.

**Reviewer Question 5**  —  In addition, a description of how CFC-12 is modeled is lacking.

**Reply**: Details about how CFC-12 was simulated will be included in the Methods section of the revised manuscript.

Minor comments

**Reviewer Question 6** — Abstract, line 10: $C_{ant}$ is not defined yet.

**Reply**: $C_{ant}$ is defined in line 5.

**Reviewer Question 7** — Page 3, line 2: a reference to the figure displaying the map of the Arctic should be made here; the reader does not necessarily know about the area characteristics. In this sense Fig. 2 should become Fig. 1.

**Reply**: The revised manuscript will include these suggested changes.

**Reviewer Question 8** — Page 3, line 2: "The bathymetry of the Arctic Ocean differs from that of the in other other oceans..."

**Reply**: In the revised manuscript we will change "of the in other other" to "in other".

**Reviewer Question 9** — Page 3, line 25: is 'laminar' right?

**Reply**: Yes, "laminar" is a common term used to describe coarse-resolution ocean models. See for example Penduff et al (2011).

**Reviewer Question 10** — Page 4, line 2: table 3 does not come into order.

**Reply**: In the revised manuscript, that table will be put in order.

**Reviewer Question 11** — Page 4, lines 11 and 12: "NEMO uses partial steps so that the model better matches the observed topography. Thus the depth of the deepest cell can be smaller than the original grid cell." Could you develop or reformulate? It is hard to understand what it is meant here.

**Reply**: We propose to rephrase the sentence as follows: "NEMO uses the partial-step approach for the model to better match the observed topography. In this approach, the bathymetry of the model is not tied directly to the bottom edge of the deepest ocean grid level, which varies with latitude and longitude; rather, the deepest ocean grid level for each column of grid cells is partially filled in to better match the observed ocean bathymetry."

**Reviewer Question 12** — Page 4, line 22: there is no mention of the Si:P and Fe:P ratios.

**Reply**: The Fe:C and Chl:C ratios of both phytoplankton groups as well as the Si:C ratio of diatoms are predicted prognostically by PISCES. These model details will be mentioned in the revised manuscript.

**Reviewer Question 13** — Page 4, line 29: does sediment mobilization only intervene in the Fe cycle? Or does it also affect the other nutrients?

**Reply**: Yes, sediment mobilization only intervenes in the Fe cycle, a point that will be clarified in the revised manuscript.

**Reviewer Question 14** — Page 4, line 30: "... following the lead of Moore et al. (2004)."

**Reply**: That phrase will be changed to "following Moore et al. (2004).".

**Reviewer Question 15** — Page 5, line 33: "... simulations made in with the same circulation model..."

**Reply**: In that sentence, "in with" will be changed to "with".

**Reviewer Question 16** — Pages 5 and 6: the many occurrences of '$xCO_2$ ' should be changed into '$CO_2$ '.

**Reply**: To avoid any confusion, we prefer to explicitly refer to '$xCO_2$ ', namely the atmospheric mixing ratio or mole fraction of $CO_2$ , i.e., the number of moles of $CO_2$ per mole of air. That ratio is typically multiplied by 106 and given in ppmv (or simply ppm) because $CO_2$ is a trace gas in the atmosphere. In the text, we need to distinguish between $xCO_2$ and the partial pressure of $CO_2$ ($pCO_2$ ), which always has pressure units ($\mu$atm). Although these two quantities are often confused, they are not the same and our method depends on keeping them straight.

**Reviewer Question 17** — Page 6, equation (1): what are the units of $pCO_2$ and T?

**Reply**: The units of for $pCO_2$ [$\mu$atm] and T [$^\circ$C] will be indicated in the revised manuscript

**Reviewer Question 18** — Page 6, line 25: "given that it is based on results from ORCA05."

**Reply**: This above-mentioned phrase was part of a paragraph that will be removed in the revised version of manuscript. That paragraph explains how the biogeochemical

**BGD**

simulations (all three resolutions) were corrected using the ORCA05 $C_{ant}$ perturbation runs. Now with our complete set of $C_{ant}$ perturbation simulations, two $C_{ant}$ perturbation tracers in each of the three resolutions as detailed earlier, this explanation is not necessary.

**Reviewer Question 19** — Page 6, line 28: reference to Table 4 should appear here.

**Reply**: This line is also part of the paragraph that will be removed in the revised manuscript.

**Reviewer Question 20** — Page 10, line 24: "apparent in ORCA025 6)."

**Reply**: The "6)" will be changed to "(Figure 6)".

**Reviewer Question 21** — Page 12, line 22: "that that excess simulated CFC-12 between 1000 and 2000 m..."

**Reply**: The double that will be changed to that.

**Reviewer Question 22** — Tables do not come into order. Table 3 should become Table 1, Table 1→2, and Table 2→3.

**Reply**: Tables will be ordered correctly in the revised manuscript.

**Reviewer Question 23** — Table 1: the 'b' subscript does not appear anywhere in the table

**Reply**: The 'b' subscript will be added to Table 1 in the revised manuscript.

**Reviewer Question 24** — Table 2, caption: "Fitted parameters for the perturbation approach for the tracers starting in 1765 (P1765) and in 1870 (P1870)."

**Reply**: In the revised manuscript, this sentence will be changed to "Fitted parameters for the perturbation simulations P1765 and P1870."

**Reviewer Question 25** — Table 4: what do exactly represent the lines "Total transport" and "Summed lateral flux"?

**Reply**: Both will be changed to 'Sum' in the revised manuscript. Both terms represent the sum of the lateral fluxes: in one case it is the lateral water flux and in the other the lateral $C_{ant}$ flux.

**Reviewer Question 26** — Table 4, caption: "Simulated values are calculated for the same time period as observations."

**Reply**: This text will be revised as proposed by the Referee.

**Reviewer Question 27** — Fig. 1 and Fig. 2 should be inverted

**Reply**: These figures will be inverted in the revised manuscript.

**Reviewer Question 28** — Fig. 10, caption: The first sentence "Profiles of $\Omega_A$ after the early industrial period period simulated only in ORCA05 (1870–1957), after intializing

the other models in 1958." is confusing. I suggest to remove most of it; it is not needed.

**Reviewer Question 29** — "Results are shown for ORCA05 in 1960 (black solid) as well as for ORCA2 (green dot- dash), ORCA05 (red dashes), and ORCA025 (blue dots) in 2012."

**Reply**: In the revised manuscript, both sentences will be simplified and combined: "Profiles of $\Omega_{arag}$ for ORCA05 in 1960 (black solid) as well as ORCA2 (green dot-dash), ORCA05 (red dashes), and ORCA025 (blue dots) in 2012."

References

Penduff, T., M. Juza, B. Barnier, J. Zika, W.K. Dewar, A. Treguier, J. Molines, and N. Audiffren, 2011: Sea Level Expression of Intrinsic and Forced Ocean Variabilities at Interannual Time Scales. J. Climate, 24, 5652–5670, https://doi.org/10.1175/JCLI-D-11-00077.1

Schlosser, P., Kromer, B., Östlund, G., Ekwurzel, B., Bönisch, G., Loosli, H., and Purtschert, R. (1994). On the 14C and 39Ar Distribution in the Central Arctic Ocean: Implications for Deep Water Formation. Radiocarbon, 36(3), 327-343. doi:10.1017/S003382220001451X

Tanhua, T., E. P. Jones, E. Jeansson, S. Jutterstrm̈, W. M. Smethie Jr., D. W. R. Wallace, and L. G. Anderson (2009), Ventilation of the Arctic Ocean: Mean ages and inventories of anthropogenic CO2 and CFC‐11, J. Geophys. Res., 114, C01002, doi: 10.1029/2008JC004868.
Waugh, D. W., Hall, T. M., McNeil, B. I., Key, R. and Matear, R. J. (2006), Anthropogenic CO2 in the oceans estimated using transit time distributions. Tellus B, 58: 376-389. doi:10.1111/j.1600-0889.2006.00222.x

---

## Author Response (AR1)

**Response to Referees**

We thank both referees for their comments and suggestions. The manuscript was revised to thoroughly address each point. Generally the plan was executed as follows:

(1) The description of the simulations analyzed in our study was improved by adding more details concerning (a) how the different simulations (ORCA2, ORCA05, and ORCA025) were made, making it clear that each of these has its own control simulation, (b) how CFC-12 was simulated, and (c) how carbon transport was estimated at the boundaries of the Arctic Ocean.

(2) The discussion section was expanded to provide more detail about (a) the effect of increasing resolution in ocean models found by other modeling studies and (b) the mechanisms influencing changes in air-sea $CO_2$ fluxes in the Arctic between the different resolutions. Additional analysis were also included to show how the different configurations compare to each other in terms of the global-ocean inventory of anthropogenic carbon.

During the review period, we also discovered an issue with our CFC-12 simulations (initialization to non-zero concentrations). Hence we have rerun all CFC-12 simulations (following details in the revised manuscript). Furthermore, we have used the opportunity to complement the ORCA05 $C_{ant}$ perturbation simulations with analogous simulations for the ORCA2 and ORCA025 configurations (each initialized in the beginning of 1958 with output from the last time step in 1957 of the ORCA05 $C_{ant}$ perturbation simulation and run until 2012). With these updated simulations, the model-data CFC-12 comparison has been improved (as discussed in Sections 3.2 and 4.1) along with the corrections for the estimated $C_{ant}$ fluxes at the boundaries (both lateral and at the air-sea interface). The figures and tables of the revised manuscript are updated accordingly. Despite these improvements, the Conclusions of our study remain the same.

In the following we address the concerns of the Reviewers point by point.
* * *
**Referee # 1**

**1) Overview**

Terhaar et al. ask: what effect does model resolution have on simulated Arctic Ocean anthropogenic $CO_2$ storage and acidification? The answer: increased model resolution shows higher regional storage by up to 25%, moving the inventory closer to data-based estimates, and increased acidification with faster shoaling of the aragonite saturation horizon. This is an interesting and useful question, and the study has been well-designed to answer it. The results and their interpretation seem sensible, although as mentioned below, a robust uncertainty analysis is critically lacking. The manuscript is interesting and easy to read. The Introduction is very well-written and Methods are clear. Results and Discussion are succinct, but in places the Discussion in particular could be developed further to provide more insight. Many of the questions below are really prompts in this direction.

**Reply**: Thank you very much. In the revised manuscript, we have strived to address these concerns in detail.

**2) Scientific questions**

**Reviewer Question 1** — The highest-resolution model is "still not eddy resolving" (3-28 i.e. page 3, line 28). Does that mean that you would expect further changes still with yet higher resolution? Would you expect the anthropogenic $CO_2$ inventory to increase even more?

**Reply**: Without actually making higher resolution simulations, one can of course never be certain. However, we would agree with Referee #1 that the anthropogenic carbon inventory in the Arctic Ocean would probably increase if the model resolution was refined even further. The $C_{ant}$ inventory in 2005 increases by 0.38 Pg C between ORCA2 and ORCA05 and by 0.27 Pg C between ORCA05 and ORCA025, so we may not have reached the nearly flat part of the asymptote. Further enhancement of resolution could further increase the inventory via the combined effects of improved bathymetry, increased coastal water velocities, and enhanced surface-to-deep transport of passive tracers from brine formation on Arctic shelves. Concerning the latter, even higher model resolution might well lead to higher $C_{ant}$ concentrations in the Canada basin and to refining the chimneys of higher CFC-12 concentrations in the Canada basin, which are observed but only barely resolved in ORCA025 (page 10, lines 23-24). We propose the following change in section 4.4 of the revised manuscript:

**Proposed change**: A signature of this source in the observed sections may be the chimneys of constant CFC-12 concentration from the surface to about 1000 m in the Canada basin, features for which only ORCA025 exhibits any such signature, albeit faint. To adequately model lateral  exchanges of $C_{ant}$ in the Arctic Ocean, at least a resolution comparable to that used in ORCA05 may be needed, while resolutions comparable to that in ORCA025 or above may well be required to begin to capture the effects from density flows along the slope. As a consequence of the deficient representation of these density flows, we would expect to see an increase in $C_{ant}$ when using even higher resolution. (page 16, lines 11–16 in the revised manuscript)

**Reviewer Question 2** — The monthly averaging process introduces an error of 27% for the lowest-resolution model (7-20). This is a similar magnitude to the difference in anthropogenic $CO_2$ inventory between the different resolutions, which is your most important result. Does this error being the same size as the 'signal' not significantly reduce your confidence in the results (i.e. differences between resolutions)?

**Reply**: Our error estimate of 27% applies only to the ORCA2 transport of $C_{ant}$ calculated separately at each of the 4 boundaries of the Arctic Ocean from monthly-average model output (offline), an important point raised by Referee #1 for which we should have been clearer. That offline calculation error is smaller at higher resolution (e.g, 4% with ORCA025). Moreover, it does not apply to our estimates of $C_{ant}$ inventory, the cumulative air-sea $C_{ant}$ flux, and the total lateral flux, all of which are calculated online in the model and for which associated calculation errors are negligible.

**Proposed change**: The relative error for transport of $C_{ant}$ across the separate boundaries introduced by the monthly average calculations is 28% for ORCA2, 7% for ORCA05, and 3% for ORCA025. Note that this error applies neither to the $C_{ant}$ inventory, nor to the cumulative air-sea flux or the lateral fluxes, which are all calculated 'online', during the simulations. (page 8, lines 9–11 in the revised manuscript)

**Reviewer Question 3** — You note that overestimation of sea-ice cover should reduce air-sea $CO_2$ exchange (9-14). There are a number of observational studies that attempt to quantify this effect. Can these be used to quantify your statement?

**Reply**: Our overestimation of sea-ice extent is less than 3%. Thus it is of second order when estimating the $C_{ant}$ air-sea flux. There are studies that have estimated the effect of declining sea-ice on air-sea fluxes of total carbon (Bates et al. [2006], Cai et al. [2010]), but they do not agree on whether the flux will increase due to increasing biological activity or decrease due to higher stratification and increases in riverine organic carbon. Because these studies focus on the air-sea flux of total $CO_2$ (natural + anthropogenic), we cannot use them to extract the response of only the anthropogenic component, the focus of our study.

**Reviewer Question 4** — Section 3.4: my impression was that the primary reason to expect model resolution to influence the anthropogenic $CO_2$ inventory was because of better representation of circulation features. Therefore the increase in lateral flux, being a function of circulation, is expected – but the simultaneous increase in the air-sea flux does not seem so intuitive. Indeed if additional anthropogenic $CO_2$ is being transported into the region from elsewhere we might expect this to increase total dissolved inorganic carbon and thus reduce net air-to-sea $CO_2$ flux. Do you have a conceptual explanation for what is driving the air-sea flux increase with resolution?

**Reply**: Two mechanisms may explain the increase in the air-sea flux with resolution: (1) higher resolution increases the amount of $C_{ant}$ that is advected into the Arctic Ocean through the Fram Strait via subsurface currents, which does not substantially affect surface $C_{ant}$ concentrations nor hence air-sea exchange of anthropogenic carbon and (2) higher resolution enhances deep-water formation in the Arctic Ocean, mainly in the Barents Sea as shown in the CFC-12 profiles (Figure 7), which reduces surface $C_{ant}$ and thus enhances the air-to-sea flux of anthropogenic carbon.

**Proposed change**: With increasing water inflow, the inflow of $C_{ant}$ is also increased. Although more $C_{ant}$ is entering the Arctic Ocean, the air-sea $C_{ant}$ flux into the Arctic Ocean increases with resolution. This apparent contradiction can be explained by two mechanism: (1) Higher resolution increases the inflow of $C_{ant}$ through the Fram Strait, which is mainly occurring in subsurface currents and therefore does not substantially impact surface $C_{ant}$ concentrations nor hence air-sea exchanges of $C_{ant}$ and (2) higher resolution enhances deep-water formation, mainly in the Barents Sea, which reduces surface $C_{ant}$ and thus enhances the air-to-sea flux of $C_{ant}$. Although the air-sea flux increases slightly, the larger lateral water fluxes in ORCA05 and ORCA025  mainly explain their higher  $C_{ant}$ concentrations in the Nansen and Amundsen basins. (page 15, lines 27–34 in the revised manuscript)

**Reviewer Question 5** — In order to declare that two things are 'not statistically different' (14-12) you must also provide the statistical information that were used to demonstrate this.

**Reply**: In the revised manuscript, the corresponding sentence is changed to 'The simulated air-sea flux falls within that assigned uncertainty range for the data-based estimate.'

**Proposed change**:  The simulated air-sea  flux falls within that assigned uncertainty range for the data-based estimate. (page 15, lines 18–19 in the revised manuscript)

**Reviewer Question 6** — Does the increase in resolution alter lateral anthropogenic $CO_2$ fluxes primarily because of the representation of circulation (1) inside the Arctic Ocean, (2) at its boundaries/interfaces, (3) in the non-Arctic global ocean, or (4) everywhere?

**Reply**: We tried to address this question in Section 4.4 of the submitted manuscript, but we only discuss circulation differences inside the Arctic Ocean and at its boundaries. We showed that increasing resolution from ORCA2 to ORCA05 affects mainly the circulation at the boundaries (1) while the change from ORCA05 to ORCA025 affects mainly circulation inside the Arctic Ocean (2).

For more insight into the role of the global ocean (outside the Arctic), we have now calculated the total $C_{ant}$ inventory for the global ocean with all three resolutions (see below). Globally the three resolutions agree within 3%. This agreement is much tighter than that for the Arctic Ocean $C_{ant}$ inventory. This new comparison suggests a weak role of the non-Arctic global ocean on the Arctic Ocean $C_{ant}$ budget (3)(4), although resolution-dependent changes in regions adjacent to the Arctic Ocean's lateral boundaries (e.g., in the North Atlantic) may well have an effect. These considerations are added to section 4.4 of the revised manuscript.

**Proposed change**: The change from ORCA2 to ORCA05 seems to mainly improve lateral exchanges with adjacent oceans, while the change from ORCA05 to ORCA025 improves inner-Arctic Ocean circulation. (page 16, lines 4–6 in the revised manuscript)

**Reviewer Question 7** — You note that for computational reasons we cannot routinely run these models globally at high resolution, but if only one region of the model needed to be at high resolution to achieve your results, would it be possible to strike a balance with a hybrid resolution model?

**Reply**: Certainly such approaches, using for example the nesting tool of NEMO (AGRIF, Debreu et al. [2008]) as in Duteil et al. [2014] would be an interesting option. Unfortunately, this nesting tool is known to not work well in ice-covered regions nor with the biogeochemical model PISCES. Hence we leave this for future work as is mentioned in the Conclusion of the revised manuscript.

**Proposed change**: For such regional studies, nested models would offer the advantage of focused higher resolution while still avoiding adverse effects from imposed lateral boundary conditions. (page 18, lines 31–32 in the revised manuscript)

**Reviewer Question 8** — Some of the notes about possible future work on CFC-12 and the TTD parameters in the Conclusions would probably be more suited to the Discussion.

**Reply**: We considered moving these notes to the Discussion, but decided against it as it is rather an outlook than a discussion of results.

**Reviewer Question 9** — No other studies have been mentioned that have attempted to answer this same question for the Arctic Ocean, but there have been other investigations of the effect of increased model resolution in various contexts. Do these provide any insights that would be useful in interpreting your results?

**Reply**: In terms of ocean biogeochemistry, most studies on the effect of resolution have focused on the impacts of mesoscale to submesoscale structures on phytoplankton and productivity (e.g., Lévy [2008],

McGillicuddy [2016]). Other studies have investigated the role of increased model resolution on transient tracers (Lachkar et al., [2007]) and on carbon and oxygen (Duteil et al., [2014]), but they have focused on other regions (Southern Ocean and the tropics). Therefore it is difficult to transfer their findings to the Arctic Ocean. Some mention on other studies focusing on the role of model resolution is included in the Discussion of the revised manuscript.

**Proposed change**:

Similar to our results in the Arctic Ocean, improving circulation with higher model resolution has also been shown to be the key driver for an improved representation of anthropogenic tracers in the Southern Ocean (Lachkar et al., 2007) or oxygen concentrations in the tropics (Duteil et al., 2014). (page 16, lines 17–19)

**3) Figures**

**Reviewer Question 10** — Use of red vs green (e.g. Fig. 3) with no difference in line style can render these lines indistinguishable to colorblind readers.

**Reviewer Question 11** — The blue-green-yellow-red color scale used on transect plots (e.g. Fig. 4) is not perceptually uniform, leading to visual artifacts such as false boundaries.

**Reviewer Question 12** — Depth should be positive going down into the ocean (Figs. 4 and 6).

**Reply**: These suggestions for improving the style and colors will be implemented in the revised manuscript.

**4) Technical/grammatical notes**

**Reviewer Question 13** — There is inconsistency in usage of past and present tenses in the Methods.

**Reply**: The Methods section will be improved to avoid this inconsistency in the revised manuscript.

**Reviewer Question 14** — The contexts in which the word 'though' is used are highly colloquial and, to me, not suited for scientific writing (2-12, 14-29, 15-8).

**Reply**: We avoid the use of the word 'though' in the revised manuscript.

**Reviewer Question 15** — Suggested corrections:
1-6    eddy-admitting
2-3    consequences for
4-30   following Moore et al.
10-24 (Fig. 6) ?
11-21 Arctic Ocean basins
15-33 reword this sentence to indicate the direction of the effect

**Reply**: The manuscript will be revised according to these suggestions. For 15-33 we will write "Thus model resolution also affects the time at which waters become undersaturated with respect to aragonite with higher resolutoin producing greater shoaling."

**Referee # 2**

In this study the authors examine the anthropogenic $CO_2$ budget of the Arctic Ocean and how this inventory depends on model resolution. In that purpose they take advantage of the NEMO-v3.2 OGCM coupled to the biogeochemical model PISCES-v1. They perform experiments with three different horizontal resolution of the OGCM, namely $2°$, $0.5°$, and $0.25°$. Inventories of anthropogenic carbon in the Arctic appear to increase with increasing resolution (from 2.0 to 2.6 Pg C). The role of air-sea fluxes and lateral transport in building these inventories is examined. In this model lateral transport accounts for 75% of the Arctic Ocean anthropogenic $CO_2$ inventory. A comparative study of the outputs of other modeling studies (CMIP5) allow concluding that models with larger lateral transport appear to better fit data-based estimates of the anthropogenic carbon in the Arctic Ocean. This partitioning does not depend on the model resolution. Resolution appears important in shaping the tracer distribution and improving data-model agreement.

The paper is well written and very well structured. However I have several concerns about the method and the way data-model comparison is performed. Before the method is thoroughly assessed this paper is not fit for publication.

**Major comments**

**Reviewer Question 1** — I have serious concerns about the applied method for estimating the $C_{ant}$. Conclusions about the impact of model resolution might not be robust due to shortcoming in the method.

$C_{ant}$ is rightly defined as the difference between the simulated historical and control ocean dissolved carbon contents. However, there is only one control experiment performed (page 5), that for ORCA05. As far as I understand the $C_{ant}$ for ORCA2 and ORCA025 is evaluated as the difference between the respective 1958→2012 experiments and the ORCA05 control for the same period. Therefore I strongly suspect that the differences in CFC and $C_{ant}$ among the different models may be explained by model drift. In order to lift that concern the following actions should be taken:

    a. Perform control experiments over the period 1958→2012 for each model resolution.

    b .While it is defensible to reduce the computation length with the high-resolution model (ORCA025) there is no such need for ORCA2, which runs even faster than ORCA05. The authors should also present results of historical and control experiments performed with ORCA2. The perturbation experiments should also be repeated with ORCA2.

    The results of these additional experiments should then be compared to those presented in the present paper. This would provide a means of validating their method and assessing potential drifts

**Reply**:

    a. We realize now that our original manuscript is woefully unclear about this point. We did not properly convey what was done. Indeed, each of our three resolutions already has its own control experiment over 1958-2012. For each resolution, $C_{ant}$ was computed from two simulations (historical and control), both made at the same resolution. Therefore, there is no resolution-related drift issue. This

point will be clarified in section 2.2 of the revised manuscript.

b. As pointed out, we had already made a control simulation in ORCA2-PISCES from 1958 to 2012 although that was not clear in the submitted manuscript. For the same period, we have in addition added a $C_{ant}$ perturbation simulation in ORCA2. Our strategy to consistently use the same ORCA05 output from the end of 1957 to initialize ORCA2 as well as ORCA025 in 1958 was by design. It produces a consistent set of results whereby the effect of resolution can be compared rigorously. Had we started all ORCA-PISCES simulations in 1870, which was not computationally feasible, there would have been larger differences due to resolution, e.g., based on the divergence shown in Figure 1. Hence the differences due to resolution discussed in the submitted manuscript are probably a lower limit, something that is clarified in the revised manuscript. In addition, for the revised manuscript, we made an additional $C_{ant}$ perturbation simulation in ORCA2 initialized in 1765 (and 1870). That new simulation provides a more complete comparison with the analogous reference ORCA05 simulation as suggested by Referee #2.

**Proposed change**:

a. Both, the control and the historical simulations, were made for all three resolutions between 1958 to 2012 to correct potential model drifts. We defined the difference between the historical simulation and the control simulation as the anthropogenic component.  While the ORCA2 and ORCA025 simulations are presented for the first time, the ORCA05 simulations were  previously used by Bourgeois et al. (2016) to assess the budget of anthropogenic carbon in the coastal ocean. (page 5, lines 21–25 in the revised manuscript)

**and**

 To account for the missing carbon, we added the difference between two perturbation simulations, one  starting in 1765 (P1765) and the other one in 1870 (P1870).  For consistency, we applied the same initialization strategy as for the biogeochemical simulations, i.e. using ORCA05 until the end of 1957 with that output serving as the initial fields for subsequent 1958–2012 simulations in all three configurations. The difference of $C_{ant}$ between P1765 and P1870 was later added to the NEMO-PISCES simulations, for each resolution separately. (page 6, lines 7–11, line 3 in the revised manuscript)

b. Lastly, we also made a perturbation simulation with using only ORCA2 from 1765 to 2012, which enables us to evaluate our simulation strategy, i.e. using ORCA05 until 1957 and then all three configurations from 1858 to 2012 (ORCA2, ORCA05, and ORCA025). (page 7, lines 1–3)

**and**

Meanwhile, we also consider that the simulated Arctic $C_{ant}$ inventory in ORCA025 may well be too low because it was initialized with ORCA05 results in 1958. Had ORCA025 been initialed instead in 1765, which was not computationally feasible, its simulated inventory would probably be larger. Although we cannot assess this affect directly, we can do so indirectly by running tests at lower resolution and noting how trends differ between model resolutions after 1958. First, let us estimate how that same 1958 initialization affects the ORCA2 results, by taking the difference in simulated $C_{ant}$ inventory between (1) the ORCA2 biogeochemical simulation from 1958 to 2012 initialized with ORCA05 in 1957 minus (2) the ORCA2 perturbation simulation from 1765 to 2012. That difference is −0.4 Pg C in 2005. Next

let us assume that there is a symmetry during 1765-2005 about the ORCA05 result with ORCA2 being lower and ORCA025 being higher as seen for the simulated period after 1958 (Figure **??** and Table **??**). We infer then that ORCA025 Arctic $C_{ant}$ inventory in 2005 would be ∼0.4 Pg C larger had it been run initialized in 1765 rather than with the ORCA05 output in 1958. If so, the ORCA025 $C_{ant}$ inventory in 2005 would increase from 2.6 to 3.0 Pg C, pushing it to closer to the center of the data-based range of 2.5–3.3 Pg C from [**?**]. After correcting both ORCA2 and ORCA025 for their 1958 initialization with ORCA05 output, the model range for the Arctic $C_{ant}$ inventory would then be 1.6–3.0 Pg C in 2005, emphasizing even more the need to go beyond coarse-resolution models in the Arctic. (page 14, lines 13–26 in the revised manuscript)

**and**

At the same time, our highest resolution inventory is likely an underestimation as it was initialized in 1958 with ORCA05 results from 1765–1957. Details in model-data based comparison differ(page 18, lines 3–4 in the revised manuscript)

**Reviewer Question 2** — The experiments, which are presented here are global. What would be the global figures for anthropogenic $CO_2$ uptake in the 5 cases? How do these figures compare to other assessments? Answering this request would allow evaluating whether the OGCMs as a whole would need serious refinements or should the effort concentrate on less-well resolved areas such as the Arctic Ocean.

**Reply**: The global corrected $C_{ant}$ inventories for 1765-2008 for the three resolutions are given below as are the uncorrected inventories for 1870-2008 (in parentheses):

- ORCA2: 152 (127) Pg C

- ORCA05: 146 (121) Pg C

- ORCA025: 148 (124) Pg C

These corrections were made by adding the difference between the two tracers in the $C_{ant}$ perturbation simulations, one initialized to zero in 1765 and the other in 1870, in each of the three resolutions. Those global perturbation results are as follows:

- ORCA2: 127 Pg C/153 Pg C

- ORCA05: 125 Pg C/150 Pg C

- ORCA025: 117 Pg C/142 Pg C

In the revised manuscript then, we present the corrected and uncorrected $C_{ant}$ inventories for the biogeochemical simulations, plus the corrections from the perturbation simulations. Thus we show results for 9 cases rather than 5 because of the new perturbation simulations in ORCA2 and ORCA025 (4 additional cases since each has 2 perturbation tracers: one initialized in 1765 and the other in 1870, each again relying on ORCA05 until 1958) that were not provided in the previously submitted version of the manuscript. We also compare our results to the calculated corrections by Bronselaer et al. (2017).

Regarding other assessments, Khatiwala et al. (2009) report a data-based estimate for the global $C_{ant}$ inventory for the period from 1765-2008 to be $140 \pm 24$ Pg C. For that same period, our results all lie within that range, falling near the upper boundary. Given the agreement of model results and

data-based estimates for the global ocean $C_{ant}$ budget, it does not appear that further enhancements to resolution are needed to improve the global carbon budget. However, we would expect that improving resolution will have a large impact on some regional budgets, e.g., in zones where the ratio between the areas of shelf seas vs. open ocean is relatively large, such as in the Arctic Ocean. Inventories of anthropogenic carbon in regions with small areal extent will have little impact on the global inventory, but they do provide some indicator of the potential enhanced effect of ocean acidification in those regions. These results and concerns will be brought up in section 4.2 of the revised manuscript.

**Proposed change**:

 Simulated global ocean $C_{ant}$ inventories are 152 Pg C in ORCA2, 146 Pg C in ORCA05, and 148 Pg C in ORCA025 in 2008, all of which account for corrections for an earlier starting date from our perturbation simulations (P1765–P1870). The corrections are similar for each resolution, e.g., 24–25 Pg C in 1995,  and are consistent with our biogeochemical model simulation strategy (all three resolutions initialized with the ORCA05 output in 1958). Furthermore, these model-based corrections are much like the $29 \pm 5$ Pg C correction calculated for the same 1765-1995 period with a data-based  approach Bronselaer et al. (2017) . For the 1765–2008 period, the data-based global $C_{ant}$ inventory estimate from Khatiwala et al. (2009) is $140 \pm 24$ Pg C, the uncertainty range of which encompasses the results from all three model resolutions. (page 11, lines 17–23 in the revised manuscript)

**and**

To adequately model lateral  exchanges of $C_{ant}$ in the Arctic Ocean, at least a resolution comparable to that used in ORCA05 may be needed, while resolutions comparable to that in ORCA025 or above may well be required to begin to capture the effects from density flows along the slope. As a consequence of the deficient representation of these density flows, we would expect to see an increase in $C_{ant}$ when using even higher resolution. (page 16, lines 13–16 in the revised manuscript)

**Reviewer Question 3** — The other main concern deals with the correction of data-based reconstructions of $C_{ant}$ (Abstract, Sections 4.2 and 4.5, Fig. 9). The authors assume that reconstructed deep values of $C_{ant}$ should be corrected downwards since observed CFC-12 concentrations at those depths are negligible. Doing so means overlooking the important fact that CFCs started to be emitted in the atmosphere much later than $CO_2$ .. Data-based estimates relying on the TTD method take into account the different tracer histories in the atmosphere. Clearly, the TTD method has limitations. The end-product displays rather large uncertainties. However, there are no sound arguments for setting the $C_{ant}$ in the deep Arctic to zero.

**Reply**: Actually, there is some evidence that the GLODAPv2 estimate using the TTD method may overestimate $C_{ant}$ concentrations in the deep Arctic Ocean. First, the water mass mean ages of deep and bottom waters are estimated to be about 250 to 300 years in the Eurasian basins (Nansen and Amundsen basins) and around 450 years in the Canadian basin (Makarov and Canada basins) (Tanhua et al., [2009]; Schlosser et al., [1994]). Thus one would expect very little if any $C_{ant}$ would have reached those old deep waters. Second, the TTD method is known to estimate $C_{ant}$ concentrations around 5 $\mu$molkg$^{-1}$ even when the CFC-12 concentrations approach zero (Waugh et al. [2006]), which demonstrates the lack of sensitivity and large uncertainty associated with the TTD estimates for older water masses. Given that data-based $C_{ant}$ concentrations in the old, deep Arctic Ocean water masses

are 18% of surface concentrations while CFC-12 concentrations in the same deep water masses are only 3% of surface concentrations, it is plausible that the TTD method overestimates $C_{ant}$.

In the revised manuscript, we rephrased the text so as to indicate that to calculate the maximum potential error in the TTD-based estimate for the $C_{ant}$ inventory in the Arctic, we set the deep TTD estimates to zero. In addition, we took out the reference to this estimation from the abstract and section 4.5.

**Proposed change**:

There are reasons to suspect that the GLODAPv2 estimate using the TTD method may overestimate $C_{ant}$ in the deep Arctic. First, the water mass mean ages below 2000 m are shown to be of the order of 300 to 400 years (Tanhua et al., 2009; Schlosser et al., 1994), older than the atmospheric $CO_2$ perturbation. Second, the TTD method estimates $C_{ant}$ concentrations ($\sim$5 $\mu$mol kg$^{-1}$), even if the CFC-12 concentrations approach zero (Waugh et al., 2006), which demonstrates the large uncertainty of the method when dealing with old water masses. To assess the maximum error associated with these potentially excessive deep TTD $C_{ant}$ estimates, we recalculated the $C_{ant}$ budget after zeroing out the $C_{ant}$ below 2000 m. Doing so reduces the data-based inventory  of $C_{ant}$ in the Arctic Ocean in 2002 by 10%. Applying the same 10% relative decrease to both the upper and lower limits of the data-based range from Tanhua et al. (2009)  leads to a minimum $C_{ant}$ inventory of 2.2– 3.0 Pg C in 2005. Simulated inventories from both ORCA05 and ORCA025 are within this lower limit. (page 14, lines 4–12 in the revised manuscript)

**and**

When the CMIP5 models are compared to the corrected data-based estimate of the  $C_{ant}$ inventory (Sect. ??), only the MIROC-ESM  with its inventory of 2.7 Pg C  fall within the data-based uncertainty estimate (2.5 to 3.3 Pg C in 2005). Nearby  is the NorESM1-ME and HadGEM2-ES, which  fall below the lower limit by 0.1 and 0.5 Pg C, receptively. Further off are the MPI-ESM and GFDL-ESM models with their  $C_{ant}$ inventories in 2005 that are 0.9 to 1.5 Pg C lower than the lower limit. The lowest estimates  are from both versions of the IPSL model whose inventories reach only $\sim$20% of the lower limit of our revised data-based range. Adjusting the CMIP5-model Arctic inventories upward by $\sim$0.4 Pg C to account for their late start date in 1850, as we did for our three simulations, would place  two of them (MIROC-ESM, and NorESM1-ME) above the lower boundary of our revised data-based uncertainty estimate, and HadGEM2-ES just 0.1 Pg C below this lower boundary. (page 16, lines 22–30 in the revised manuscript)

**and**

We replaced as well the corrected estimate by the uncorrected estimate in Figure 9.

**Reviewer Question 4** — Modeled CFC-12 inventories in the Arctic (Fig. 5 and page 12, lines 24 and 25) appear to be much lower than the observed ones, even with ORCA25. Would it be possible to provide total (integrated over depth and distance) inventories along the AOS94 and Beringia 2005 expedition pathways and compare the 3 model results to the data inventories? The low CFC inventory provides an indication that low $C_{ant}$ would be expected too.

**Reply**: Excellent suggestion. Thank you. We have now calculated these total inventories integrated over depth and distance along sections:

- Beringia 2005

    - Observations: 9.4 $\mu$mol m$^{-1}$
    - ORCA025: 7.7 $\mu$mol m$^{-1}$
    - ORCA05: 5.8 $\mu$mol m$^{-1}$
    - ORCA2: 3.7 $\mu$mol m$^{-1}$

- AOS94

    - Observations: 5.5 $\mu$mol m$^{-1}$
    - ORCA025 : 4.8 $\mu$mol m$^{-1}$
    - ORCA05 : 3.5 $\mu$mol m$^{-1}$
    - ORCA2 : 2.0 $\mu$mol m$^{-1}$

For both expeditions, the observed CFC-12 section inventories are underestimated by 13-18% in ORCA05, 31-38% in ORCA05, and 61-64% in ORCA2. This tendency with resolution for these section inventories is consistent with that seen for the Arctic Ocean's basin-wide inventory where the data-based estimate is underestimated by 13-15% in ORCA025, 22-24% in ORCA05, and 34-36% in ORCA2.

**Proposed change**: Lastly, we calculate CFC-12 inventories along the two sections, integrated over depth and distance (Table 6). Depending on the expedition, ORCA025 underestimates the observed CFC-12 section inventories by 13-18%, ORCA05 by 31-38%, and ORCA2 by 61-64%. (page 11, lines 13–15)
>    **and**
>    A table (table 6) of these integrated values is added.

**Reviewer Question 5** — In addition, a description of how CFC-12 is modeled is lacking.

**Reply**: Details about how CFC-12 was simulated is included in the Methods section of the revised manuscript.

**Proposed change**: CFC-12 is a purely anthropogenic tracer, a sparingly soluble gas whose concentration began to increase in the atmosphere in the early 1930's, part of which has been transferred to the ocean via air-sea gas exchange. Its uptake and redistribution in the ocean has been simulated following OCMIP-2 protocols (Dutay et al., 2002). The CFC-12 flux ($F_{CFC}$) at the air-sea interface was calculated as follows:

$$F_{CFC} = k_w(\alpha_{CFC}\ p\text{CFC} - C_s)(1 - I), \tag{1}$$

where $k_w$ is the gas-transfer velocity (piston velocity) in m s$^{-1}$ (Wanninkhof, 1992), $p$CFC the atmospheric partial pressure of CFC-12 in atm from the reconstructed atmospheric history by Bullister (2015), $C_s$ is the sea surface concentration of CFC-12 (mol m$^{-3}$), $\alpha_{CFC}$ is the solubility of CFC-12 (mol m$^{-3}$atm$^{-1}$) from Warner and Weiss (1985), and $I$ is the model's fractional sea-ice cover. Once

**Minor comments**

**Reviewer Question 6** — Abstract, line 10: $C_{ant}$ is not defined yet.

**Reply**: $C_{ant}$ is defined in line 5.

**Reviewer Question 7** — Page 3, line 2: a reference to the figure displaying the map of the Arctic should be made here; the reader does not necessarily know about the area characteristics. In this sense Fig. 2 should become Fig. 1.

**Reply**: The revised manuscript includes these suggested changes.

**Reviewer Question 8** — Page 3, line 2: "The bathymetry of the Arctic Ocean differs from that of the in other other oceans..."

**Reply**: In the revised manuscript we changed "of the in other other" to "in other".

**Reviewer Question 9** — Page 3, line 25: is 'laminar' right?

**Reply**: Yes, "laminar" is a common term used to describe coarse-resolution ocean models. See for example Penduff et al (2011).

**Reviewer Question 10** — Page 4, line 2: table 3 does not come into order.

**Reply**: In the revised manuscript, that table is put in order.

**Reviewer Question 11** — Page 4, lines 11 and 12: "NEMO uses partial steps so that the model better matches the observed topography. Thus the depth of the deepest cell can be smaller than the original grid cell." Could you develop or reformulate? It is hard to understand what it is meant here.

**Reply**: We propose to rephrase the sentence as follows: "NEMO uses the partial-step approach for the model to better match the observed topography. In this approach, the bathymetry of the model is not tied directly to the bottom edge of the deepest ocean grid level, which varies with latitude and longitude; rather, the deepest ocean grid level for each column of grid cells is partially filled in to better match the observed ocean bathymetry."

**Reviewer Question 12** — Page 4, line 22: there is no mention of the Si:P and Fe:P ratios.

**Reply**: The Fe:C and Chl:C ratios of both phytoplankton groups as well as the Si:C ratio of diatoms are predicted prognostically by PISCES. These model details are mentioned in the revised manuscript.

**Reviewer Question 13** — Page 4, line 29: does sediment mobilization only intervene in the Fe cycle? Or does it also affect the other nutrients?

**Reply**: Yes, sediment mobilization only intervenes in the Fe cycle, a point that is clarified in the revised manuscript.

**Reviewer Question 14** — Page 4, line 30: "... following the lead of Moore et al. (2004)."

**Reply**: That phrase was changed to "following Moore et al. (2004).".

**Reviewer Question 15** — Page 5, line 33: "... simulations made in with the same circulation model..."

**Reply**: In that sentence, "in with" was changed to "with".

**Reviewer Question 16** — Pages 5 and 6: the many occurrences of 'xCO$_2$ ' should be changed into 'CO$_2$ '.

**Reply**: To avoid any confusion, we prefer to explicitly refer to 'xCO$_2$ ', namely the atmospheric mixing ratio or mole fraction of CO$_2$ , i.e., the number of moles of CO$_2$ per mole of air. That ratio is typically multiplied by $10^6$ and given in ppmv (or simply ppm) because CO$_2$ is a trace gas in the atmosphere. In the text, we need to distinguish between xCO$_2$ and the partial pressure of CO$_2$ (pCO$_2$ ), which always has pressure units ($\mu$atm). Although these two quantities are often confused, they are not the same and our method depends on keeping them straight.

**Reviewer Question 17** — Page 6, equation (1): what are the units of pCO$_2$ and T?

**Reply**: The units of pCO$_2$ [$\mu$atm] and T [$^\circ$C] are indicated in the revised manuscript

**Reviewer Question 18** — Page 6, line 25: "given that it is based on results from ORCA05."

**Reply**: This above-mentioned phrase was part of a paragraph that will be removed in the revised version of manuscript. That paragraph explains how the biogeochemical simulations (all three resolutions) were corrected using the ORCA05 C$_{ant}$ perturbation runs. Now with our complete set of C$_{ant}$ perturbation simulations, two C$_{ant}$ perturbation tracers in each of the three resolutions as detailed earlier, this explanation is unnecessary.

**Reviewer Question 19** — Page 6, line 28: reference to Table 4 should appear here.

**Reply**: This line is also part of the paragraph that is removed in the revised manuscript.

**Reviewer Question 20** — Page 10, line 24: "apparent in ORCA025 6)."

**Reply**: The "6)" is changed to "(Figure 6)".

**Reviewer Question 21** — Page 12, line 22: "that that excess simulated CFC-12 between 1000 and 2000 m..."

**Reply**: The double that is changed to that.

**Reviewer Question 22** — Tables do not come into order. Table 3 should become Table 1, Table 1→2, and Table 2→3.

**Reply**: Tables are ordered correctly in the revised manuscript.

**Reviewer Question 23** — Table 1: the 'b' subscript does not appear anywhere in the table

**Reply**: The 'b' subscript is added to Table 1 in the revised manuscript.

**Reviewer Question 24** — Table 2, caption: "Fitted parameters for the perturbation approach for the tracers starting in 1765 (P1765) and in 1870 (P1870)."

**Reply**: In the revised manuscript, this sentence is changed to "Fitted parameters for the perturbation simulations P1765 and P1870."

**Reviewer Question 25** — Table 4: what do exactly represent the lines "Total transport" and "Summed lateral flux"?

**Reply**: Both are changed to 'Sum' in the revised manuscript. Both terms represent the sum of the lateral fluxes: in one case it is the lateral water flux and in the other the lateral $C_{ant}$ flux.

**Reviewer Question 26** — Table 4, caption: "Simulated values are calculated for the same time period as observations."

**Reply**: This text is revised as proposed by the Referee.

**Reviewer Question 27** — Fig. 1 and Fig. 2 should be inverted

**Reply**: These figures are inverted in the revised manuscript.

**Reviewer Question 28** — Fig. 10, caption: The first sentence "Profiles of $\Omega_A$ after the early industrial period period simulated only in ORCA05 (1870–1957), after intializing the other models in 1958." is confusing. I suggest to remove most of it; it is not needed.

**Reviewer Question 29** — "Results are shown for ORCA05 in 1960 (black solid) as well as for ORCA2 (green dot- dash), ORCA05 (red dashes), and ORCA025 (blue dots) in 2012."

**Reply**: In the revised manuscript, both sentences are simplified and combined: "Profiles of $\Omega_{arag}$ for ORCA05 in 1960 (black solid) as well as ORCA2 (green dot-dash), ORCA05 (red dashes), and ORCA025 (blue dots) in 2012."

[revised manuscript text omitted]

---

## Referee Report (RR1)

**Comments on "Model constraints on the anthropogenic carbon budget of the Arctic Ocean"**

by Terhaar et al., revised version, submitted to Biogeosciences, doi: 10.5194/bg-2018-283

First of all I appreciate that the authors carefully considered most of previous comments. As I already mentioned the paper is clear and provides interesting results.

However, some concerns remain in addition to some errors in the revised version. Were these concerns lifted, the paper would then be fit for publication.

1. In my first review I had questioned the adequacy of the initialization with ORCA05 results in 1958 and suggested that this method be compared for ORCA2 with an experiment identical to that performed with ORCA05 over the entire period. I expected results of this sensitivity experiment to be presented in Figures 2, 7, and 9 and discussed in the text. However, the authors only performed a simplified perturbation simulation with ORCA2. This does not allow assessing the method consisting in the initialization of ORCA2 and ORCA025 with ORCA05 results in 1958. In addition, except for the mention in Section 4.2, there is no discussion of that additional experiment.

2. The discussion of model data comparison (Section 4.2) is mostly speculative and relies on a misunderstanding of the TTD method.

    The fact that models predict lower values than data-based $C_{ant}$ reconstructions is no proof that reconstructions overestimate $C_{ant}$ as sentence on lines 2-3 page 14 suggests. The different model versions clearly underestimate CFC-12 invasion (Sections 3.2 and 4.1). In consequences one would also expect $C_{ant}$ to be underestimated. In that respect why do the authors insist on lowering data-based estimates?

    While the TTD method as applied by Tanhua et al. (2009) or in GLODAP-v2 is less well constrained for large transit times (since this method relies on tracers with a short atmospheric history) it does not follow that $C_{ant}$ should be set to zero whenever CFC concentrations are very low. Any water parcel in the ocean is characterized by a distribution of transit times (TTD) which differs from a delta-function due to the presence of mixing (e.g., Waugh et al., 2006). The mean of that distribution corresponds to the mean water age and its width depends on mixing strength and pathways. The assumption that the TTD width is equal to its mean seems to be adequate enough for most ocean areas  (Waugh et al., 2006; Tanhua et al., 2009). Hence the water body under consideration is characterized by ages ranging from zero to the mean age and beyond. Taking that into account, and acknowledging that CFCs and CO2 do not have the same atmospheric history (CFC-12 concentrations in the atmosphere started to rise significantly after 1950 while the anthropogenic carbon perturbation started 2 centuries earlier) a mean age of 300 to 400 years does not preclude any $C_{ant}$ contribution at depth.

    Additionally there is no rationale for assuming that "there is a symmetry during 1765-2005 about the ORCA05 result with ORCA2 being lower and ORCA025 being higher" even if it happens to be the case after 1958. There are many processes at stake (air-sea exchange, lateral transport, mixing, atmospheric increase…). Therefore the response of the system is

expected to be non-linear. In consequences there is no justification for assuming that the $C_{ant}$ inventory with ORCA025 would be larger by 0.4 PgC had the experiment started in 1765.

Further, the ORCA2 experiment which result in a lower inventory is not a biogeochemical experiment but a perturbation one which relies on a simplified carbon cycle. Inventories should not be corrected on that basis. Results of a complete biogeochemical experiment with ORCA2 starting in 1765 would be needed for such assessment. I am rather surprised that no such experiment seems to be available?

Rather than aiming at reconciling modeled $C_{ant}$ and data-based reconstruction this section should be devoted to discussing $C_{ant}$ in view of the CFC-12 results.

3.  Conclusions need to be revised along the preceding lines (Page 17 lines 29-32 and page 18,line 1-4). The conclusions should also mention that all model versions underestimate CFC inventories in the Arctic, hence underestimated $C_{ant}$ inventories.

4.  Model vertical resolution and mixing schemes.

    a.  In the model description on page 4 the authors state

        "Vertically, all three model configurations have the same discretization, where the full-depth water column is divided into 46 depths levels, whose thicknesses increase from 6 m at the surface to 500 m in the deepest grid box"

        If I am well informed the 500 m box thickness for deep boxes is typical of ORCA2 with 31 levels while a thickness of 6 m at the surface is typical of the 46 levels versions.

        Is the vertical grid spacing actually the same for all 3 model versions?

    b.  Additionally, one may wonder if the vertical diffusivity and viscosity are represented the same way in all 3 versions? Could the authors add information on that aspect?

**Miscellaneous**

*   The CFC model-data misfits quoted on page 11 (lines 13-15) do not agree with the values in Table 6. In the later ORCA2 displays the best agreement with data!

*   Figure 4: the bottom right panel displays CFC-12 results and not temperature.

*   Page 7, line 2: "...using ORCA05 until 1957 and then all three configurations from  **1958** to 2012..."

*   Page 11, line 13: Table 6 does not come into order; should be Table 5

*   Page 11, line 29: "In 2002, the upper limit  of the modeled $C_{ant}$ inventory range..."

*   Page 11,line 32: " **This** correction is 0.4 Pg C in 2005 for each resolution..."

*   Page 12, line 25: "… flow fields is 0.05 Pg C (~3%)  smaller than..."

*   Page 14, line 15: "Although we cannot assess this  **effect** directly,"

---

## Author Response (AR2)

**Response to the Editor**

Dear Prof. Marilaure Grégoire,

We have carefully addressed the comments from Referee #2 in the revised manuscript and the attached point-by-point response.

As requested, we have now made the full biogeochemical simulation in the ORCA2 model as requested by Referee #2. Furthermore, we have also thoroughly modified Section 4.2 to carefully address their criticism.

In addition, we include a new Appendix demonstrating that the perturbation approach agrees with the full approach within 3%, and we include a short paragraph at the beginning of the Methods section to provide a general overview of the simulations that were made.

Thank you for the effort that you have put into evaluating and improving this work.

Sincerely,

Jens Terhaar

**Response to Referee # 2**

We thank Referee #2 for their comments. Below is a point-by-point response, which accompanies a thoroughly revised manuscript.
* * *
**Referee # 2**

First of all I appreciate that the authors carefully considered most of previous comments. As I already mentioned the paper is clear and provides interesting results. However, some concerns remain in addition to some errors in the revised version. Were these concerns lifted, the paper would then be fit for publication.

**Major comments**

**Reviewer Question 1** — In my first review I had questioned the adequacy of the initialization with ORCA05 results in 1958 and suggested that this method be compared for ORCA2 with an experiment identical to that performed with ORCA05 over the entire period. I expected results of this sensitivity experiment to be presented in Figures 2, 7, and 9 and discussed in the text. However, the authors only performed a simplified perturbation simulation with ORCA2. This does not allow assessing the method consisting in the initialization of ORCA2 and ORCA025 with ORCA05 results in 1958. In addition, except for the mention in Section 4.2, there is no discussion of that additional experiment.

**Reply**:  As requested, we have now made the full biogeochemical simulation in ORCA2, i.e. the one that is analogous to that performed with ORCA05 (1870–2012). The results are now presented in Figures 2, 7, and 9. These results are discussed in sections 3.3 and 4.5. Section 4.2 has been thoroughly modified.

**Reviewer Question 2** — The discussion of model data comparison (Section 4.2) is mostly speculative and relies on a misunderstanding of the TTD method.
    The fact that models predict lower values than data-based $C_{ant}$ reconstructions is no proof that reconstructions overestimate Cant as sentence on lines 2-3 page 14 suggests. The different model versions clearly underestimate CFC-12 invasion (Sections 3.2 and 4.1). In consequences one would also expect Cant to be underestimated. In that respect why do the authors insist on lowering data-based estimates?
    While the TTD method as applied by Tanhua et al. (2009) or in GLODAP-v2 is less well constrained for large transit times (since this method relies on tracers with a short atmospheric history) it does not follow that $C_{ant}$ should be set to zero whenever CFC concentrations are very low. Any water parcel in the ocean is characterized by a distribution of transit times (TTD) which differs from a delta-function due to the presence of mixing (e.g., Waugh et al., 2006). The mean of that distribution corresponds to the mean water age and its width depends on mixing strength and pathways. The assumption that the TTD width is equal to its mean seems to be adequate enough for most ocean areas (Waugh et al., 2006; Tanhua et al., 2009). Hence the water body

under consideration is characterized by ages ranging from zero to the mean age and beyond. Taking that into account, and acknowledging that CFC's and $CO_2$ do not have the same atmospheric history (CFC-12 concentrations in the atmosphere started to rise significantly after 1950 while the anthropogenic carbon perturbation started 2 centuries earlier) a mean age of 300 to 400 years does not preclude any $C_{ant}$ contribution at depth.

**Reply**: In Section 4.2, we have removed any speculation and no longer attempt to adjust the data-based estimates downward for an improved model-data comparison. We clearly state that our models underestimate $C_{ant}$ concentrations in the deep waters of the Arctic Ocean, while emphasizing the importance of the CFC-12 model evaluation for that conclusion. In the revised version of section 4.2, we also mention that the different data-based approaches that have been used to estimate $C_{ant}$ generally yield different results in deep waters and that the TTD approach tends to be on the high end (Khatiwala et al., 2013).

Additionally there is no rationale for assuming that "there is a symmetry during 1765–2005 about the ORCA05 result with ORCA2 being lower and ORCA025 being higher" even if it happens to be the case after 1958. There are many processes at stake (air-sea exchange, lateral transport, mixing, atmospheric increase...). Therefore the response of the system is expected to be non-linear. In consequences there is no justification for assuming that the $C_{ant}$ inventory with ORCA025 would be larger by 0.4 Pg C had the experiment started in 1765.

**Reply**: We have now removed the quantitative argument, based on symmetry, that the Arctic Ocean inventory in ORCA025 would have been 0.4 Pg C higher had that simulation started in 1765 instead of being initialized in 1958 with results from ORCA05. We now emphasize simply the direction of the expected change, an increase based on (1) the greater penetration of CFC-12 into intermediate waters in ORCA025 relative to ORCA05 (Figure 5) and (2) the weaker penetration of CFC-12 for ORCA2 along with the corresponding lower simulated $C_{ant}$ inventory in ORCA2 when it was integrated since 1765 compared to it being initialized in 1958 with results from ORCA05. Moreover, the sign of the divergence of ORCA2 from ORCA05 is always consistent (negative), whether initialized from ORCA05 in 1958 or in the two perturbation simulations (initialized in 1765 and in 1870). The same consistency must hold for ORCA025 relative to ORCA05, although the sign of its divergence is opposite.

In regards to the physical processes that are mentioned by Referee #2, they are nonlinear, but there is no substantial trend nor shift in trend in the physical forcing that was imposed on the model throughout the simulation. That is, the DFS forcing has little trend as pointed out in the revised manuscript.

In summary, all of our transient tracer simulations indicate that the ventilation of subsurface waters is stronger in ORCA025 than in ORCA05. The former absorbs more CFC-12 and $C_{ant}$ than the latter. It follows that any "partial" simulation where ORCA025 is initialized with ORCA05 output partway through, as we have done due to computational limitations, will absorb less $C_{ant}$ than a full ORCA025 simulation. We see no way to escape this logic. Moreover it is supported by the consistent divergence of ORCA2 from ORCA05 in three simulations (initialized at different times along the way), all in the opposite direction since subsurface ventilation in ORCA2 is weaker than in ORCA05.

Further, the ORCA2 experiment which result in a lower inventory is not a biogeochemical experiment but a perturbation one which relies on a simplified carbon cycle. Inventories should not be corrected on that basis. Results of a complete biogeochemical experiment with ORCA2 starting in 1765 would be needed for such assessment. I am rather surprised that no such experiment seems

to be available?

**Reply**: We have now made an ORCA2 biogeochemical simulation starting in 1870, identical to the ORCA05 simulation to test the effect of branching. It would not have been consistent to start that new full biogeochemical simulation in ORCA2 in 1765 because the full biogeochemical simulation in ORCA05 only started in 1870.

The assertion by Referee #2 that inventories should not be corrected based on the perturbation approach is founded on the assumption that it gives very different results than does the full biogeochemcial approach. Because a rigorous comparison of the two approaches has never been published, we now offer this comparison as an Appendix in the revised manuscript. It shows that the $C_{ant}$ inventory simulated with the perturbation approach agrees within 3% of that from the full biogeochemical approach, both globally and regional. Moreover, there is remarkable overall structural agreement. Therefore, we still use the difference between two perturbation simulations (one starting in 1765 and another in 1870) to correct for the late starting date (1870) of the full biogeochemical simulations, each made in a consistent fashion at the different resolutions (ORCA025, ORCA05, ORCA2, and ORCA2$^\star$).

Rather than aiming at reconciling modeled $C_{ant}$ and data-based reconstruction this section should be devoted to discussing $C_{ant}$ in view of the CFC-12 results.

**Reply**: As mentioned above, Section 4.2 no longer attempts to reconcile model and data-based $C_{ant}$ by adjusting the deep-water, data-based estimates downward. It now offers a more refined discussion of $C_{ant}$ in light of the CFC-12 results while also mentioning some literature that discusses uncertainties associated with the data-based $C_{ant}$ estimates in deep waters.

**Reviewer Question 3** — Conclusions need to be revised along the preceding lines (Page 17 lines 29-32 and page 18,line 1-4). The conclusions should also mention that all model versions underestimate CFC inventories in the Arctic, hence underestimated $C_{ant}$ inventories.

**Reply**: As in section 4.2, the revised Conclusions no longer mention any downward adjustment of the TTD data-based estimates. They also now mention that all model configurations underestimate $C_{ant}$ inventories because they generally underestimate CFC-12.

**Reviewer Question 4** — Model vertical resolution and mixing schemes.

a. In the model description on page 4 the authors state "Vertically, all three model configurations have the same discretization, where the full- depth water column is divided into 46 depths levels, whose thicknesses increase from 6 m at the surface to 500 m in the deepest grid box" If I am well informed the 500 m box thickness for deep boxes is typical of ORCA2 with 31 levels while a thickness of 6 m at the surface is typical of the 46 levels versions. Is the vertical grid spacing actually the same for all 3 model versions?

**Reply**: There was confusion on this point because we did not mention in this sentence that the "partial steps" formulation can essentially double the thickness of the the bottom layer. In section 2.1 of the revised manuscript we eliminate this confusion, writing the following: "Vertically, all three model configurations have the same discretization, where the full-depth water column is divided into 46 levels

whose thicknesses vary from 6 m (top level) to 249 m (level 45), but that the latter can reach up to 498 m, being extended into level 46 as a function of the bathymetry (partial steps)."

b. Additionally, one may wonder if the vertical diffusivity and viscosity are represented the same way in all 3 versions? Could the authors add information on that aspect?

**Reply**: We have now added the following sentence to section 2.1: "Vertically, the same eddy viscosity $(1.2 \times 10^{-4} \text{ m}^2 \text{ s}^{-1})$ and diffusivity coefficients $(1.2 \times 10^{-5} \text{ m}^2 \text{ s}^{-1})$ were used in all three resolutions."

**Minor comments**

**Reviewer Question 5** — The CFC model-data misfits quoted on page 11 (lines 13-15) do not agree with the values in Table 6. In the later ORCA2 displays the best agreement with data!

**Reply**: The labels for ORCA2 and ORCA025 were reversed and have been corrected.

**Reviewer Question 6** — Figure 4: the bottom right panel displays CFC-12 results and not temperature.

**Reply**: That erroneous label has been corrected.

**Reviewer Question 7** — Page 7, line 2: "...using ORCA05 until 1957 and then all three configurations from  1958 to 2012..."

**Reply**: This correction has been made.

**Reviewer Question 8** — Page 11, line 13: Table 6 does not come into order; should be Table 5

**Reply**: Table 5 and Table 6 have now been put into the proper order.

**Reviewer Question 9** — Page 11, line 29: "In 2002, the upper limit  of the modeled Cant inventory range..."

**Reply**: Done

**Reviewer Question 10** — Page 11,line 32: " This correction is 0.4 Pg C in 2005 for each resolution..."

**Reply**: Done

**Reviewer Question 11** — Page 12, line 25: "... flow fields is 0.05 Pg C ( 3%)  smaller than..."

**Reply**: Done

**Reviewer Question 12** — Page 14, line 15: "Although we cannot assess this  effect directly"

**Reply**: Done

[revised manuscript text omitted]